# Beyond Instance-Level Alignment: Dual-Level Optimal Transport for Audio-Text Retrieval

**Wenqi Guo**[1], **Shikui Tu**[1*], **Lei Xu**[1,2*]
[1]Shanghai Jiao Tong University, Shanghai, China
[2] Guangdong Laboratory of Artificial Intelligence and Digital Economy (SZ), Shenzhen, China
{wenqiguo, tushikui, leixu}@sjtu.edu.cn

## Abstract

Cross-modal matching tasks have achieved significant progress, yet remain limited by mini-batch subsampling and scarce labelled data. Existing objectives, such as contrastive losses, focus solely on instance-level alignment and implicitly assume that all feature dimensions contribute equally. Under small batches, this assumption amplifies noise, making alignment signals unstable and biased. We propose DART (Dual-level Alignment via Robust Transport), a framework that augments instance-level alignment with feature-level regularization based on the Unbalanced Wasserstein Distance (UWD). DART constructs reliability-weighted marginals that adaptively reweight channels according to their cross-modal consistency and variance statistics, highlighting stable and informative dimensions while down-weighting noisy or modality-specific ones. From a theoretical perspective, we establish concentration bounds showing that instance-level objectives scale with the maximum distance across presumed aligned pairs, while feature-level objectives are governed by the Frobenius norm of the transport plan. By suppressing unmatched mass and sparsifying the transport plan, DART reduces the effective transport diameter and tightens the bound, yielding greater robustness under small batches. Empirically, DART achieves state-of-the-art retrieval performance on three audio-text benchmarks, with particularly strong gains under scarce labels and small batch sizes.

## 1 Introduction

Audio-text retrieval is a fundamental cross-modal matching task that supports applications in multimedia search (Elizalde et al., 2019) and content understanding (Oncescu et al., 2024). The key challenge lies in learning aligned representations that capture semantic correspondences between heterogeneous modalities, enabling the retrieval of audio clips given text queries and vice versa. Existing approaches, including learn-to-match frameworks (Luong et al., 2024; Shi et al.; Li et al., 2019), contrastive learning (Jia et al., 2021; Radford et al., 2021; Mei et al., 2022; Wu et al., 2023), and triplet losses (Wei et al., 2021; Zeng et al., 2022), can be viewed under a unified inverse optimal transport (IOT) perspective (Shi et al., 2023), where paired supervision is used to learn a shared metric between audio and text features.

In practice, this metric is optimized from mini-batches. As the batch size decreases, sampling variance increases, amplifying feature fluctuations and making the learned metric more susceptible to noise and bias. One key reason is the reliance on instance-level similarity, typically measured by Euclidean distance or cosine similarity. Such measures collapse each audio-text pair into a single scalar and implicitly treat all embedding dimensions as equally informative. However, audio and text embeddings are heterogeneous: stable semantic cues often coexist with modality-specific noise and transient patterns across dimensions. Such uniform collapse allows a few noisy channels to dominate the scalar score, yielding unstable alignment signals and biased gradients even for semantically matched pairs. Prior channel-weighting methods (for example, Luong et al., 2024) mitigate this

---
∗Corresponding authors

by rescaling dimensions, but they still reduce the weighted embeddings to a single pairwise score, leaving volatile channels coupled to the learning signal, particularly under small-batch training.

This observation motivates moving beyond purely instance-level alignment. To mitigate instability and bias, we propose **DART** (Dual-level Alignment via Robust Transport), which augments instance-level alignment with feature-level regularization. At the instance level, DART adopts an IOT objective to enforce tight alignment between paired audio and text samples. At the feature level, DART treats each embedding channel as a matching unit and minimizes the Unbalanced Wasserstein Distance (UWD) between audio and text features. Noisy channels tend to incur larger transport costs, which leads UWD to assign them less mass and thus suppress spurious alignment, while stable semantic channels with smaller costs are preferentially matched. This intuition is supported by our empirical analysis in Appendix (Fig. 2), where injecting synthetic noise into feature channels yields a monotonic increase in their standardized transport cost. Beyond this implicit filtering, DART introduces Reliability-Aware Marginals (RAM) as priors in UWD. For each channel, it computes a reliability score based on variance, kurtosis, and cross-modal correlation. The scores are normalized into reliability-aware marginals in UWD, steering the transport plan toward high-reliability channels that consistently encode stable semantic cues, while downweighting volatile or modality-specific ones.

From a theoretical perspective, we establish that instance-level and feature-level alignments exhibit fundamentally different concentration behaviors. For standard instance-level alignment, the mini-batch objective aggregates pairwise similarities, and we prove that its concentration bound scales with the maximum distance ($D_{\max}$) among positive pairs in the batch. This reliance on an extremal, worst-case quantity explains the sensitivity of vanilla IOT to outlier samples and noisy labels. In contrast, our proposed feature-level formulation yields bounds governed by the Frobenius norm of the optimal transport plan ($\|\mathbf{P}^*\|_F$). By shifting the controlling quantity from a volatile maximum distance to an aggregate norm, DART provides tighter guarantees and greater robustness, particularly in small-batch regimes where $D_{\max}$ is prone to high variance. Overall, our contributions are threefold:

- We introduce DART, a dual-level alignment framework that augments instance-level IOT with structural feature-level regularization, enabling robust cross-modal retrieval by simultaneously optimizing sample and channel relationships.

- We design reliability-aware marginals that incorporate statistical cues (variance, kurtosis, correlation) to reweight feature channels, effectively suppressing modality-specific noise within the transport process.

- We provide a theoretical analysis connecting feature-level alignment to tighter concentration bounds. Extensive experiments demonstrate that DART achieves state-of-the-art performance on three benchmarks, showing significant gains under challenging small-batch and limited-label conditions.

## 2 PRELIMINARIES

### 2.1 ENTROPIC OPTIMAL TRANSPORT AND INVERSE OPTIMAL TRANSPORT.

Entropic optimal transport (EOT) extends classical OT by adding an entropy regularization term, which improves computational efficiency and yields smooth couplings (Cuturi, 2013). Given empirical measures $\mu$ and $\nu$, EOT solves

$$\min_{\mathbf{\Pi} \in U(\mu, \nu)} \langle \mathbf{C}, \mathbf{\Pi} \rangle - \epsilon H(\mathbf{\Pi}), \tag{1}$$

where $\mathbf{C} \in \mathbb{R}^{m \times n}$ is the ground cost matrix ($C_{ij} = d(x_i, y_j)$), $H(\mathbf{\Pi}) = -\sum_{i,j} \Pi_{ij}(\ln \Pi_{ij} - 1)$ is the negative entropy, and $\epsilon > 0$ is the regularization parameter. The feasible set $U(\mu, \nu) = \{\mathbf{\Pi} \in \mathbb{R}_+^{m \times n} \mid \mathbf{\Pi}\mathbf{1}_n = \mu, \mathbf{\Pi}^\top \mathbf{1}_m = \nu\}$ enforces the marginal constraints.

In cross-modal retrieval, the ground cost is often unknown and must be learned. Inverse optimal transport (IOT) (Dupuy et al., 2016; Li et al., 2019; Stuart & Wolfram, 2020) parameterizes the cost as $\mathbf{C}^\theta$ (e.g., via a neural network) and optimizes $\theta$ such that the induced coupling $\mathbf{\Pi}^\theta$ aligns with

the observed ground-truth matching $\widetilde{\mathbf{\Pi}}$:

$$\min_{\theta} KL\big(\widetilde{\Pi} \,\|\, \Pi^{\theta}\big), \quad \text{where} \quad \mathbf{\Pi}^{\theta} = \arg\min_{\mathbf{\Pi} \in U(\mu,\nu)} \left\langle \mathbf{C}^{\theta}, \mathbf{\Pi} \right\rangle - \epsilon H(\mathbf{\Pi}), \tag{2}$$

where $KL(\cdot\|\cdot)$ denotes the Kullback-Leibler divergence, and $\mathbf{f}, \mathbf{g}$ are the dual potentials obtained from the Sinkhorn iterations.

## 2.2 AUDIO-TEXT RETRIEVAL AS IOT.

Audio-text retrieval aims to align audio clips with their corresponding text captions. Given audio and caption data pairs $\mathcal{D} = \{(x_i, y_i)\}_{i=1}^n$, where $x_i$ represents the $i$-th audio sample and $y_i$ the associated text caption, the goal is to learn a cross-modal mapping that enables bidirectional retrieval between audio queries and text captions. Specifically, audio samples are encoded via an audio encoder $f_{\theta}(\cdot)$, while text captions are encoded via a text encoder $g_{\phi}(\cdot)$. A distance function $d(\cdot, \cdot)$, such as Euclidean or cosine distance, is employed to measure the similarity between the audio and text embeddings, denoted as $\mathbf{C}_{ij} = d(f_{\theta}(x_i), g_{\phi}(y_j))$.

In the IOT framework, the network is optimized to minimize these distances for semantically matched pairs while maximizing them for unmatched ones, effectively treating the pairwise distance matrix as a parameterized ground cost. During training, the EOT solver (Eq. 1) generates a coupling matrix $\mathbf{\Pi}^{(\theta,\phi)}$ that encodes the inferred matching relationships. The model is optimized by minimizing the KL divergence between the induced coupling $\mathbf{\Pi}^{(\theta,\phi)}$ and an observed ground-truth matching $\widetilde{\mathbf{\Pi}}$, where $\widetilde{\mathbf{\Pi}}_{ij} = 1/n$ if $i = j$ and 0 otherwise. This objective encourages the model to assign low transport costs to semantically matched pairs while penalizing misalignments.

At inference, given a set of audio samples $X_{\text{test}} = \{x_i\}_{i=1}^{m'}$ and caption samples $Y_{\text{test}} = \{y_j\}_{j=1}^{n'}$, the retrieval of a caption $\hat{y}$ for a query audio $x_i$ is performed by selecting the sample that yields the minimum distance (or maximum similarity):

$$\hat{y} = \arg\min_{y_j \in Y_{\text{test}}} \frac{\exp\left(d\left(f_{\theta}\left(x_i\right), g_{\phi}\left(y_j\right)\right)\right)}{\sum_{k=1}^{m'} \exp\left(d\left(f_{\theta}\left(x_i\right), g_{\phi}\left(y_k\right)\right)\right)}. \tag{3}$$

## 2.3 LIMITATIONS OF INSTANCE-LEVEL IOT.

While the IOT perspective provides a unifying view, it reveals critical limitations in practice. In mini-batch training, the cost matrix is estimated from partial data and aggregates all embedding dimensions uniformly. This uniform pooling ignores the inherent heterogeneity of audio and text embeddings. For example, for the caption "A drone is whirring", the embedding typically encodes meaning in a distributed manner: some channels (or directions) contribute more to object-identity cues such as drone, while others are more informative about sound-attribute cues such as the whirring acoustic pattern. n practice, while many latent channels effectively represent these stable semantic components, others inevitably encode modality-specific noise or unstable variations.

Concretely, the instance-level distance $d(x_i, y_j)$ collapses these diverse sub-components through a uniform summation (e.g., $d(x_i, y_j) = \sum_d (x_{id} - y_{jd})^2$). Because of this mandatory aggregation, even a small subset of noisy or high-variance channels can disproportionately inflate the distance for semantically matched pairs. As a result, the learned metric becomes biased toward spurious fluctuations rather than true semantic alignment. This intuition is formalized in Section 4, where Theorem 1 shows that the concentration bound for the instance-level IOT loss is governed by the maximum alignment distance $D_{\max} = \max_{(i,j): \widetilde{\Pi}_{ij} > 0} d(x_i, y_j)$. Consequently, the learning signal is dominated by these inflated outlier pairs, especially in small-batch regimes.

# 3 DART: DUAL-LEVEL ALIGNMENT VIA ROBUST TRANSPORT

## 3.1 MINI-BATCH INSTANCE-LEVEL IOT

Given a mini-batch of $k$ audio-text pairs $(X^b, Y^b)$, the encoders $f_{\theta}$ and $g_{\phi}$ produce embeddings $\mathbf{U}^b \in \mathbb{R}^{k \times d_u}$ and $\mathbf{V}^b \in \mathbb{R}^{k \times d_v}$. The cost matrix is defined as

$$\mathbf{C}_{\text{Sample}}^{(\theta,\phi)b}[i, j] = d(\mathbf{U}_{i,:}^b, \mathbf{V}_{j,:}^b), \tag{4}$$

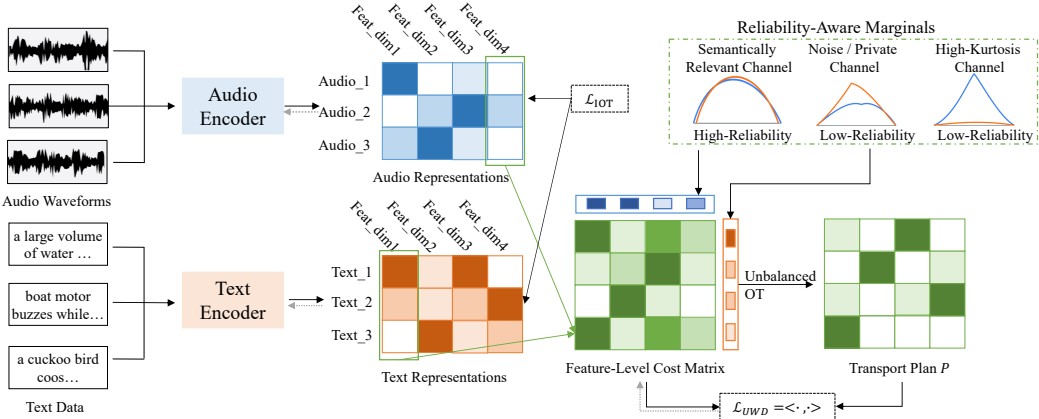

Figure 1: An overview of the proposed DART framework. DART aligns audio and text modalities through both instance-level optimization using the Inverse Optimal Transport (IOT) objective and feature-level optimization via channel-wise distribution alignment. The latter minimizes the Unbalanced Wasserstein Distance (UWD) with reliability-aware marginals to guide the transport plan toward stable semantic channels while suppressing noisy or modality-specific ones.

where $d(\cdot, \cdot)$ is a distance metric (Euclidean in our implementation). An entropy-regularized OT solver (Sinkhorn) produces a coupling $\mathbf{\Pi}^{(\theta,\phi)b}$, and the IOT objective minimizes the divergence to the ground-truth matching:

$$\mathcal{L}_{\text{IOT}}^b(\theta, \phi) = KL\big(\widetilde{\mathbf{\Pi}}^b \,\|\, \mathbf{\Pi}^{(\theta,\phi)b}\big). \tag{5}$$

which reduces to $-\log \Pi_{ii}^{(\theta,\phi)b}$ under one-to-one alignment.

This mini-batch IOT formulation is widely used and provides a baseline for retrieval tasks. However, it estimates costs from partial data and aggregates all embedding dimensions uniformly, which makes it sensitive to batch variance and noisy channels. We therefore extend IOT with feature-level optimization, as detailed next.

## 3.2 FEATURE-LEVEL DISTRIBUTION ALIGNMENT

**Audio and Text Feature-Level Representations.** In DART, each feature dimension is treated as an independent distribution across the mini-batch and aligned across the two modalities. Given the audio feature matrix $\mathbf{U}^b$ and text feature matrix $\mathbf{V}^b$, the $j$-th column of these matrices is interpreted as a distribution of the $j$-th feature across the mini-batch samples. Specifically, for the audio and text modality:

$$\mathbf{U}^b(:, j) = \begin{bmatrix} f_\theta\left(x_1^b\right)_j \\ \vdots \\ f_\theta\left(x_k^b\right)_j \end{bmatrix}, \mathbf{V}^b(:, j) = \begin{bmatrix} g_\phi\left(y_1^b\right)_j \\ \vdots \\ g_\phi\left(y_k^b\right)_j \end{bmatrix}. \tag{6}$$

Here, $f_\theta\left(x_i^b\right)_j$ denotes the value of the $j$-th feature dimension for the $i$-th audio sample. Similarly, on the text side, each feature dimension $j$ corresponds to a $k$-dimensional vector representing the distribution of this feature across the mini-batch samples.

**Ground Metric (Feature-Level).** The Wasserstein distance has become a widely adopted metric for measuring the discrepancy between probability distributions, as it considers both distributional shifts and underlying geometric structures Panaretos & Zemel (2019). Leveraging this property, DART promotes alignment at the feature level between audio and text modalities via optimal transport. Specifically, let $\mathbf{U}^b \in \mathbb{R}^{k \times d_u}$ and $\mathbf{V}^b \in \mathbb{R}^{k \times d_v}$ denote the audio and text feature matrices for

the $b$-th mini-batch of size $k$, with $d_u$ and $d_v$ as their respective feature dimensions. DART constructs a feature cost matrix $\mathbf{C}_{\text{Feature}}^{(\theta,\phi)b} \in \mathbb{R}^{d_u \times d_v}$, whose $(i,j)$-th entry measures the Euclidean distance between the distributions of the $i$-th audio feature dimension and the $j$-th text feature dimension within that mini-batch:

$$\mathbf{C}_{\text{Feature}}^{(\theta,\phi)b}[i,j] = \left\| \mathbf{U}^b(:,i) - \mathbf{V}^b(:,j) \right\|_2. \tag{7}$$

Here, $\mathbf{U}^b(:,i)$ and $\mathbf{V}^b(:,j)$ are the $k$-dimensional vectors corresponding to the $i$-th and the $j$-th features of audio and text, respectively, across the samples in the $b$-th mini-batch.

**(Unbalanced) Wasserstein Distance between Feature Distributions in a Mini-Batch.** In many real-world scenarios, feature distributions across modalities (e.g., audio and text) are inherently misaligned due to noise, missing data, and variations in feature quality or scale. Such discrepancies become more pronounced in randomly sampled mini-batches, where the total "mass" and support of the distributions may differ across modalities. Consequently, as required by the traditional Wasserstein distance, may result in suboptimal alignment. To alleviate this, DART utilizes the unbalanced Wasserstein distance (UWD), which relaxes the mass-conservation constraint by allowing mass "leakage".

Formally, given the cost matrix $\mathbf{C}_{\text{Feature}}^{(\theta,\phi)b}$, the UWD is the optimal value of the following optimization problem, with $\mathbf{P}^b$ as the transport plan:

$$\mathbf{P}^b = \arg\min_{\mathbf{P}^b \in \mathbb{R}_+^{d_u \times d_v}} \left[ \left\langle \mathbf{C}_{\text{Feature}}^{(\theta,\phi)b}, \mathbf{P}^b \right\rangle + \tau \left( KL(\mathbf{P}^b \mathbf{1}_{d_u} \| \boldsymbol{u}_{d_u}^b) + KL((\mathbf{P}^b)^T \mathbf{1}_{d_v} \| \boldsymbol{v}_{d_v}^b) \right) \right], \tag{8}$$

where $\langle \cdot, \cdot \rangle$ denotes the Frobenius inner product, and $\mathbf{1}_{d_u}(\mathbf{1}_{d_v})$ is a vector of ones of length $d_u(d_v)$. The first term, $\left\langle \mathbf{C}_{\text{Feature}}^{(\theta,\phi)b}, \mathbf{P}^b \right\rangle$, represents the overall transport cost, capturing how dissimilar the audio and text feature distributions are within the mini-batch. The second term adds a KL-based regularization that penalizes discrepancies between the marginals of the transport plan $\mathbf{P}^b$ and the uniform distributions $\boldsymbol{u}_{d_u}^b$ and $\boldsymbol{v}_{d_v}^b$. The parameter $\tau$ controls the trade-off between minimizing the cost and maintaining mass consistency.

**Mini-Batch Feature-Level Loss.** Once $\mathbf{P}^b$ is obtained, the feature-level UWD loss within the $b$-th mini-batch is defined by the total transport cost:

$$\mathcal{L}_{\text{UWD}}^b(\theta, \phi) = \left\langle \mathbf{C}_{\text{Feature}}^{(\theta,\phi)b}, \mathbf{P}^b \right\rangle. \tag{9}$$

## 3.3 Reliability-Aware Marginals (RAM)

To further guide feature-level transport toward stable semantic channels and away from volatile or modality-specific ones, DART builds reliability-aware marginals that act as priors in the unbalanced OT objective, steering mass allocation toward informative features and reducing the influence of noisy or unstable dimensions.

**Channel Reliability Estimation.** Given audio and text embeddings $\mathbf{U}^b \in \mathbb{R}^{k \times d}$ and $\mathbf{V}^b \in \mathbb{R}^{k \times d}$ for a mini-batch of size $k$, the reliability of the $j$-th channel is estimated from three complementary statistics:

$$r_j = \sigma\Big( \text{corr}(\mathbf{U}^b(:,j), \mathbf{V}^b(:,j)) - \text{var}\big(\mathbf{U}^b(:,j), \mathbf{V}^b(:,j)\big) - \text{kurt}\big(\mathbf{U}^b(:,j), \mathbf{V}^b(:,j)\big) \Big), \tag{10}$$

where $\text{corr}$ denotes normalized cross-modal correlation, $\text{var}$ captures variance instability, and $\text{kurt}$ measures heavy-tailedness. $\sigma(\cdot)$ is the sigmoid function. A higher score $r_j \in (0,1)$ indicates that channel $j$ is more likely to capture stable cross-modal semantics. Detailed definitions of these statistics and their computation are provided in Appendix B.

**Normalization into Marginals.** The reliability scores are normalized into probability distributions:

$$\boldsymbol{u}^b = \frac{\mathbf{r}}{\sum_j r_j}, \quad \boldsymbol{v}^b = \frac{\mathbf{r}}{\sum_j r_j}, \tag{11}$$

where $\mathbf{r} = (r_1, \ldots, r_d)$ is the vector of channel reliabilities. These marginals replace the uniform ones in the UWD formulation, biasing the transport plan toward reliable channels.

**Reliability-Aware UWD Loss.** Substituting the marginals into the UWD formulation yields the reliability-aware feature-level loss:

$$\mathcal{L}_{\text{UWD-R}}^b(\theta, \phi) = \min_{\mathbf{P}^b \geq 0} \langle \mathbf{C}_{\text{Feature}}^{(\theta,\phi)b}, \mathbf{P}^b \rangle + \tau \Big[ KL(\mathbf{P}^b \mathbf{1}_d \,\|\, \boldsymbol{u}^b) + KL((\mathbf{P}^b)^\top \mathbf{1}_d \,\|\, \boldsymbol{v}^b) \Big]. \quad (12)$$

Here the KL terms penalize deviations of the transport marginals from the reliability priors $(\boldsymbol{u}^b, \boldsymbol{v}^b)$. As a result, channels with higher reliability scores receive larger marginal mass, encouraging $\mathbf{P}^b$ to allocate more transport to them. This reduces the overall cost term $\langle \mathbf{C}_{\text{Feature}}, \mathbf{P}^b \rangle$, effectively lowering the feature-level loss and constraining the solution toward semantically stable dimensions while suppressing noisy ones.

**Stabilization via EMA.** To prevent fluctuations in reliability estimation from small batches, DART aggregates per-channel scores across distributed workers and updates them using exponential moving average (EMA). Specifically, for each channel $j$, the global reliability score $r_j^{(t)}$ at step $t$ is updated as

$$r_j^{(t)} = \beta r_j^{(t-1)} + (1 - \beta)\hat{r}_j^{(t)}, \quad (13)$$

where $\hat{r}_j^{(t)}$ is the score from the current mini-batch and $\beta \in (0, 1)$ is the smoothing coefficient, which we set to 0.9 in all experiments. This EMA update ensures that transient spikes or drops in small batches do not immediately affect the marginals.

DART then integrates this reliability-aware UWD loss into the overall training objective to encourage cross-modal alignment. The total loss is given by:

$$\mathcal{L}_{\text{total}} = \min_{\theta, \phi} \frac{1}{B} \sum_{b=1}^{B} \big( \mathcal{L}_{\text{IOT}}^b(\theta, \phi) + \lambda \mathcal{L}_{\text{UWD-R}}^b(\theta, \phi) \big), \quad (14)$$

where $\mathcal{L}_{\text{IOT}}^b(\theta, \phi)$ is the loss defined in 5, and $\lambda$ is a hyperparameter that balances the two losses.

## 4 CONCENTRATION BOUNDS FOR $\mathcal{L}_{\text{IOT}}$ AND $\mathcal{L}_{\text{UWD}}$

**Theorem 1** (Concentration of Instance-Level IOT Loss). *Let $\delta \in (0, 1)$ and $m$ be the fixed mini-batch size. Suppose the log function is $L$-Lipschitz on $[\epsilon, 1]$ and the optimal transport plan $\mathbf{\Pi}$ satisfies $\mathbf{\Pi}_{ij} \in [\epsilon, 1]$. Define the* maximum alignment distance *over the ground-truth support $\widetilde{\mathbf{\Pi}}$ as*

$$D_{\max} = \max_{(i,j):\, \widetilde{\mathbf{\Pi}}_{ij} > 0} d(x_i, y_j), \quad (15)$$

*namely the largest distance among audio-text pairs labeled as matches. Then, with probability at least $1 - \delta$:*

$$\big| \mathcal{L}_{\text{IOT}}^B - \mathcal{L}_{\text{IOT}}^* \big|^2 \leq \frac{\epsilon L^2}{2} \Big( D_{\max} + \epsilon(2 \log_2(m) + 1) \Big) \sqrt{\frac{\log(2/\delta)}{2B}}, \quad (16)$$

*where $B$ is the number of training batches.*

**Theorem 2** (Concentration of Feature-Level UWD Loss). *Let $\delta \in (0, 1)$ and consider the feature-level UWD loss $\mathcal{L}_{\text{UWD}}$ in equation 9. Suppose the mini-batch cost matrix $\mathbf{C}_{\text{Feature}}^{(\theta,\phi)b} \in \mathbb{R}^{d_u \times d_v}$ is estimated from $m$ i.i.d. paired samples, with variance bounded by $\sigma^2$.*

*Then, with probability at least $1 - \delta$:*

$$\big| \mathcal{L}_{\text{UWD}}^B - \mathcal{L}_{\text{UWD}}^* \big| \leq \|\mathbf{P}^*\|_F \cdot \epsilon_m + \frac{1}{2\tau} \epsilon_m^2, \quad (17)$$

*where $\epsilon_m = \sqrt{\frac{2\sigma^2 \log(2/\delta)}{m}}$, $\tau$ is the regularization parameter in equation 8, and $\mathbf{P}^*$ is the optimal feature-level transport plan.*

Table 1: Retrieval performance on AudioCaps (AuC) and Clotho (Clo) datasets. All methods are trained with a batch size of 256, consistent with the settings reported in the original papers, except for the models in the second block (rows with CNN/BPE encoders), where the batch size is reduced to 6 due to GPU memory constraints. For DART variants, *w/ RAM* denotes using reliability-aware marginals, while *w/o RAM* reduces to uniform marginals.

| Method | Encoder | T $\rightarrow$ A (AuC) | | A $\rightarrow$ T (AuC) | | T $\rightarrow$ A (Clo) | | A $\rightarrow$ T (Clo) | |
|---|---|---|---|---|---|---|---|---|---|
| | | R@1 | R@10 | R@1 | R@10 | R@1 | R@10 | R@1 | R@10 |
| (Oncescu et al., 2021) | | 28.1 | 79.0 | 33.7 | 83.7 | 9.6 | 40.1 | 10.7 | 40.8 |
| (Mei et al., 2022) | Audio: | 33.9 | 82.6 | 39.4 | 83.9 | 14.4 | 49.9 | 16.2 | 50.2 |
| (Deshmukh et al., 2022) | ResNet38 | 33.1 | 80.3 | 39.8 | 84.6 | 15.8 | 49.9 | 17.4 | 54.3 |
| (Wu et al., 2023) | Text: | 36.7 | 83.2 | 45.3 | 87.7 | 12.0 | 43.9 | 15.7 | 51.3 |
| (Luong et al., 2024) | BERT | 39.10 | 85.78 | 49.94 | 90.49 | 16.65 | 52.84 | 22.10 | 56.74 |
| DART w/o RAM | | 40.20 | 85.45 | 54.44 | **90.59** | **17.30** | 53.35 | 22.48 | 57.03 |
| DART w/ RAM | | **41.67** | **85.97** | **55.27** | 90.38 | 17.18 | **54.52** | **23.54** | **58.85** |
| (Wang et al., 2023) | A: CNN | **33.72** | **83.59** | 39.14 | 82.24 | 16.63 | 51.98 | 20.47 | 55.50 |
| DART w/o RAM | T: BPE | 33.12 | 81.93 | **43.30** | **84.11** | 19.67 | 57.18 | 26.50 | **63.25** |
| DART w/ RAM | | 33.42 | 82.53 | **43.30** | **84.11** | **20.07** | **59.08** | **26.79** | 62.00 |
| (Chen et al., 2023) | A: Beats | 54.2 | 91.2 | 66.9 | **96.7** | 36.7 | 74.4 | 25.9 | 64.7 |
| DART w/o RAM | T: BERT | 56.2 | **93.2** | 71.1 | **97.3** | 37.0 | **75.9** | 27.5 | 68.9 |
| DART w/ RAM | | **56.9** | **93.2** | **72.1** | 97.0 | **37.5** | **75.9** | **27.9** | **69.5** |

The two bounds highlight a key distinction between instance-level and feature-level formulations. For the instance-level loss in Theorem 1, the deviation is controlled by the largest alignment distance $D_{\max}$ among audio-text pairs labeled as matches. Because mini-batches only contain a restricted subset of samples, their feasible matching set is limited. When the correct partner of a sample is absent from the batch (e.g., due to label noise), the transport plan may be forced to assign mass to a higher-cost alternative. This inflates the effective $D_{\max}$ in mini-batch training compared to the global dataset, leading to a looser concentration bound and larger variance in gradient estimates.

In contrast, the feature-level bound in Theorem 2 depends on the Frobenius norm of the transport plan $\|\mathbf{P}^*\|_F$. This term measures the squared sum of all transport assignments across channels, so the deviation is controlled by the overall mass distribution rather than dominated by a single worst-case pair. As a result, occasional noisy or high-cost channels contribute only marginally to the bound, while the majority of stable semantic channels reduce the effective variance. This aggregation effect makes the bound inherently tighter and less sensitive to outliers, thereby providing greater robustness under small batches or noisy labels.

## 5 EXPERIMENTS

We evaluate the effectiveness and generalization of DART on audio-text retrieval benchmarks and beyond. We compare DART against standard baselines including contrastive learning (Radford et al., 2021; Jia et al., 2021), triplet losses (Wei et al., 2021), and OT-based methods (Shi et al., 2023). All methods are trained under the same conditions unless otherwise noted. Detailed implementation settings are provided in Appendix D.

**DART consistently enhances overall audio-text retrieval performance.** We present DART on the AudioCaps and Clotho datasets, comparing them with state-of-the-art methods using the R@1 and R@10 metrics. To ensure a fair comparison, we categorize the baselines based on their audio/text encoder architectures and adopt identical model settings, including batch sizes, for each group. As shown in Tab. 1, DART consistently superior or comparable performance across all encoder settings. For instance, with ResNet38+BERT encoders on AudioCaps, DART outperforms the strongest baseline Luong et al. (2024) by 4.5% (A→T) and 1.1% (T→A) in R@1. Similar gains are observed on Clotho, where DART leads in both R@1 and R@10. Despite matching the ONE-PEACE's Wang et al. (2023) constrained batch size of 2 (required due to model scale), DART achieves superior performance in 5 of 8 key metrics while maintaining comparable results in others.

Table 2: Retrieval performance on the AudioCaps dataset under varying semi-supervised and noisy conditions. The top rows show semi-supervised settings with 20% and 40% unlabeled data, while the bottom rows represent noisy correspondence settings with 20% and 40% of captions replaced by unrelated ones. All methods use a batch size of 32.

| Condition | Method | Text → Audio | | | Audio → Text | | |
|---|---|---|---|---|---|---|---|
| | | R@1 | R@5 | R@10 | R@1 | R@5 | R@10 |
| Semi-Supervised (20% Unlabeled) | Triplet loss | 15.34 | 48.34 | 66.88 | 24.29 | 52.83 | 69.84 |
| | Contrastive loss | 28.58 | 65.55 | 81.50 | 35.63 | 68.42 | 80.36 |
| | (Luong et al., 2024) | 32.93 | 67.43 | 80.89 | 39.81 | 70.53 | 82.44 |
| | DART | **34.85** | **70.44** | **83.34** | **45.03** | **76.28** | **86.62** |
| Semi-Supervised (40% Unlabeled) | Triplet loss | 0.1 | 0.52 | 1.06 | 0.1 | 0.52 | 1.46 |
| | Contrastive loss | 20.58 | 53.96 | 70.72 | 27.37 | 58.72 | 75.21 |
| | (Luong et al., 2024) | 28.58 | 62.69 | 77.19 | 35.00 | 69.27 | 79.72 |
| | DART | **33.24** | **69.55** | **82.74** | **43.67** | **74.39** | **87.46** |
| Noisy Labels (20% Noisy) | Triplet loss | 16.82 | 46.39 | 62.71 | 19.64 | 46.39 | 59.77 |
| | Contrastive loss | 25.80 | 61.56 | 78.16 | 33.33 | 66.66 | 78.78 |
| | (Luong et al., 2024) | 31.32 | 67.11 | 80.48 | 38.35 | 73.77 | 84.85 |
| | DART | **32.87** | **67.77** | **81.06** | **43.57** | **73.98** | **86.72** |
| Noisy Labels (40% Noisy) | Triplet loss | 0.58 | 1.58 | 2.13 | 1.14 | 4.91 | 8.98 |
| | Contrastive loss | 22.23 | 55.90 | 72.76 | 26.95 | 59.03 | 73.24 |
| | (Luong et al., 2024) | 26.20 | 61.31 | 76.17 | 34.37 | 65.30 | 77.84 |
| | DART | **29.67** | **65.30** | **80.20** | **37.09** | **67.18** | **80.45** |

**DART remains robust under small batches and noisy or semi-supervised labels.** We first evaluate DART's performance under noisy and semi-supervised conditions on the AudioCaps dataset. Noise is introduced by randomly replacing text captions with unrelated ones at ratios of 20% and 40%, while semi-supervised settings simulate scenarios where a portion of the data lacks labels entirely by randomly masking parts of the label information (in $\widetilde{\Pi}$). In this experiment, we set a small batch size of 32 to test DART's performance under limited negative samples and noisy data conditions. The results in 2 show that DART maintains stable retrieval performance even with reduced negative samples and noisy inputs, demonstrating its resilience to input perturbations. This robustness in challenging settings highlights DART's capacity to generalize well even when faced with noisy data and limited label availability. These findings underscore DART's suitability for large-scale, real-world applications where data quality and label availability may be limited, and computational resources are constrained.

**Zero-Shot Sound Event Detection.** We evaluate DART's generalization ability by conducting zero-shot sound event detection on the ESC-50 dataset. Models are pretrained on the AudioCaps dataset for the audio-caption matching task and applied directly to ESC-50 without additional fine-tuning. Following Luong et al. (2024), all classes in the test set are converted to template captions, such as "This is a sound of class." As shown in 4, we report the R@1, R@5, R@10, and mAP scores for models trained with three types of IOT loss under different constraints: triplet loss, contrastive loss, matching loss (as used in Luong et al. (2024)), and our proposed DART, with a consistent batch size of 128 for all models. DART achieves the highest R@1 score of 80.75%, outperforming triplet loss (71.25%), contrastive loss (72.25%), and matching loss (79.25%). It also shows competitive performance in R@5 and R@10, closely matching the results of Luong et al. (2024). This demonstrates DART's superior generalization to unseen sound events in a zero-shot setting. Notably, the matching loss in Luong et al. (2024) is similar to our $\mathcal{L}_{\text{IOT}}$ in 5, and the improvements highlight how the feature-level $\mathcal{L}_{\text{UWD}}$ in DART enhances alignment between audio and text distributions, boosting performance.

**DART introduces negligible GPU memory overhead compared to instance-level baselines.** A potential concern is whether feature-level transport increases GPU memory consumption. For a

batch size of $k$ and feature dimension $d{=}512$, the feature-level cost matrix $\mathbf{C}\text{Feature}^{(\theta,\phi)b}$ involves computing $d^2 = 512^2$ pairwise distances, each costing $O(k)$ operations. Storing all intermediate results in float32 requires only a few megabytes: the embedding matrices $\mathbf{U}^b, \mathbf{V}^b \in \mathbb{R}^{k\times512}$ occupy about 64KB each when $k{=}32$, and both the cost matrix $\mathbf{C}\text{Feature}$ and the transport plan $\mathbf{P}^b$ require roughly 1MB each. The unbalanced Wasserstein loss $\mathcal{L}\text{UWD}$ in Eq. (16) is computed as a point-wise product $\langle \mathbf{C}\text{Feature}, \mathbf{P}^b \rangle$, requiring no additional buffers. Importantly, $\mathbf{P}^b$ in Eq. (15) can be computed on CPU via offloaded OT solvers and is detached from the gradient graph, so during backpropagation only $\mathbf{C}_\text{Feature}$ contributes gradients. In practice, this means DART introduces only $\sim$2MB of extra GPU memory with no additional GPU cost for optimizing $\mathbf{P}^b$. Moreover, reliability estimation (variance, kurtosis, and cross-modal correlation) can be precomputed or updated offline, further ensuring that DART fits within the same GPU memory budget as instance-level IOT methods. For extremely high-dimensional encoders (e.g., $d > 2048$), one can further ensure scalability by first projecting features to a lower dimension $d' \leq 1024$ with a lightweight linear layer before computing feature-level OT, or by applying low-rank approximations of the cost matrix (such as Nyström-type methods) to reduce the effective quadratic dependence on $d$.

**DART generalizes effectively to other cross-modal tasks such as image-text retrieval.** Beyond audio-text retrieval, we evaluate DART on the MSCOCO dataset for image-text matching. As shown in Tab. 3, DART consistently improves both image→text and text→image retrieval compared to strong baselines. This demonstrates that the proposed dual-level

Table 3: Image-text retrieval performance on MSCOCO dataset. Results are reported in R@1.

| Method | Image→Text | Text→Image |
|---|---|---|
| (Shi et al.) | 19.15 | 20.90 |
| DART (ours) | **21.27** | **23.34** |

alignment and reliability-aware marginals are not tied to the audio-text domain, but transfer naturally to other heterogeneous modalities. The results highlight DART's potential as a general solution for cross-modal matching tasks.

**Ablation Study.** We conduct ablation studies to understand the contribution of each component in DART.

First, the dual-level objective is necessary. As shown in Appendix 10, using only the feature-level loss $\mathcal{L}_\text{UWD}$ fails to recover cross-modal correspondences (R@1 $\approx$ 0), whereas the instance-level IOT loss $\mathcal{L}_\text{IOT}$ provides a standard baseline. Jointly optimizing both yields the best performance, confirming their complementary roles: $\mathcal{L}_\text{IOT}$ anchors global sample alignment, while $\mathcal{L}_\text{UWD}$ acts as a structural regularizer to filter noisy feature directions.

Second, Reliability-Aware Marginals (RAM) are both effective and interpretable. Replacing RAM with uniform marginals ("DART w/o Reliability" in Tab. 1) consistently degrades retrieval accuracy, indicating that treating all channels equally is suboptimal. Further decomposition of RAM into its core variants (Table 5) reveals three key insights: (i) Correlation alone is unstable: While simple cross-modal correlation (*corr*) may slightly improve one retrieval direction, it often hurts the other (e.g., A→T R@1 drops from 51.52% to 50.05%), leading

Table 4: The zero-shot sound event detection on the ESC50 test set, the $R@1$ score is equivalent to accuracy.

| Loss | Audio → Sound | | | |
|---|---|---|---|---|
| | R@1 | R@5 | R@10 | mAP |
| Triplet | 71.25 | 91.75 | 95.75 | 80.09 |
| Contrastive | 72.25 | 93.00 | 96.75 | 80.84 |
| IOT | 79.25 | **97.5** | 99.25 | 87.09 |
| DART | **80.75** | 97.25 | **99.75** | **87.78** |

to a lower mean R@1 than the uniform baseline. This suggests that correlation is susceptible to spurious signals in mini-batches. (ii) Stability statistics are crucial: Both EMA variance (*emavar*) and kurtosis (*kurt*) provide consistent gains over the uniform baseline. They serve as effective stabilizers by down-weighting high-variance channels and penalizing heavy-tailed, outlier-dominated dimensions. (iii) The full RAM design is robust: it achieves the best overall performance (mean R@1: 45.55) with the highest A→T R@1 (52.56) while keeping T→A competitive, demonstrating that all three statistics are necessary to safeguard against different types of noise while maintaining peak accuracy.

Finally, sensitivity analyses confirm that DART is robust to variations in key hyperparameters. Finally, we study the sensitivity to key hyperparameters. Tab. 12 shows that DART yields consistent

Table 5: Core RAM variants on AudioCaps (ResNet38–BERT, batch size 64).

| Marginal Design | A→T R@1 | A→T R@10 | T→A R@1 | T→A R@10 | Mean R@1 |
|---|---|---|---|---|---|
| uniform (w/o RAM) | 51.52 | 90.80 | 38.31 | 85.77 | 44.92 |
| corr (correlation) | 50.05 | 90.60 | 38.64 | 85.22 | 44.35 |
| emavar (EMA variance) | 51.83 | 90.49 | 38.52 | 85.56 | 45.18 |
| kurt (kurtosis) | 51.93 | 90.60 | 38.64 | 85.74 | 45.29 |
| RAM (full) | **52.56** | 90.60 | 38.54 | 85.56 | **45.55** |

gains across mini-batch sizes and is particularly beneficial in the small-batch regime (e.g., clear R@1 improvements at $k=8$ and $k=32$), while remaining competitive at larger batches. Tab. 7 further indicates that performance is stable under moderate changes of the loss weight $\lambda$ (0.1–0.7), with only minor variation in retrieval metrics. Alternative marginal choices for UWD (Tab. 8) exhibit similar trends, confirming that DART is robust to reasonable design variations.

## 6 RELATED WORKS

Cross-modal matching is a fundamental challenge in multi-modal learning, aiming to establish meaningful correspondences between two modalities, such as text-image (Jia et al., 2021; Radford et al., 2021; Wei et al., 2020), text-audio (Wu et al., 2023; Deshmukh et al., 2022), by aligning their underlying distributions. Recent advancements have focused on leveraging metric learning techniques to learn joint embedding spaces where semantically similar instances from different modalities are mapped close to each other, using methods such as triplet loss (Mei et al., 2022; Wei et al., 2020), contrastive learning (Radford et al., 2021; Yang et al., 2022), and matching loss (Shi et al.). Despite their success, these approaches treat all embedding dimensions equally, implicitly assuming that each channel contributes similarly to semantic alignment. In practice, however, many dimensions may be noisy, redundant, or modality-specific, making uniform treatment suboptimal. Although Luong et al. (2024) introduces per-channel coefficients that reweight feature dimensions, their formulation only assigns weights to corresponding dimensions across modalities (e.g., audio $j$-th dimension with text $j$-th dimension), which effectively assumes one-to-one channel alignment. Such a constraint overlooks potential cross-channel correspondences (e.g., an audio rhythm dimension aligning better with a textual verb-related dimension), thereby limiting flexibility. Moreover, their method still relies heavily on sample-level supervision, leaving it vulnerable to small-batch variance and noisy labels.

## 7 CONCLUSION

We presented DART, a dual-level alignment framework for audio-text retrieval that couples instance-level IOT with a feature-level regularizer based on unbalanced optimal transport, targeting the sensitivity of instance-level objectives to feature-level noise and mini-batch sampling variance. DART moves beyond treating embeddings as holistic units: it treats embedding channels as matchable units and, through reliability-aware marginals, concentrates transport mass on channels that consistently encode stable semantic cues while downweighting volatile or modality-specific ones. From a theoretical standpoint, we analyzed the mini-batch instability of instance-level alignment and derived concentration bounds showing a worst-positive dependence on the maximum paired distance, whereas the feature-level formulation is controlled by the Frobenius norm of the transport plan, yielding tighter guarantees under small batches and noisy supervision. Empirically, DART improves retrieval accuracy on three benchmarks across encoder choices, with pronounced gains in small-batch settings, and it also transfers to image-text retrieval, suggesting that the feature-level view is not tied to a specific modality. Beyond performance, DART provides an interpretable mechanism for robustness: the transport plan reveals which channels are emphasized, and RAM offers a lightweight way to inject stability priors without introducing heavy additional networks. This indicates that feature-level regularization is not merely a heuristic, but a principled complement to instance-level matching when channel-wise noise can dominate scalar similarities.

ACKNOWLEDGEMENTS

This work was supported by the Science and Technology Commission of Shanghai Municipality (24510714300), and National Natural Science Foundation of China (62172273).

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

## LLM USAGE STATEMENT

We used large language models (e.g., ChatGPT) as general-purpose writing assistants to improve the readability and clarity of the manuscript. The research ideas, methodology, experimental design, and analysis were conceived and carried out entirely by the authors.

## APPENDIX OVERVIEW

The appendix is organized into the following sections with additional analysis, proofs, and implementation details:

- In Section A, we provide full training pseudocode of DART.

- In Section B, we detailed implementation of the reliability-aware marginal estimation module.

- In Section C, we present theoretical analysis, including notation, mini-batch sampling paradigms, and concentration bounds for $\mathcal{L}_{\text{IOT}}$ and $\mathcal{L}_{\text{UWD}}$.

- In Section D, we describe implementation details such as hardware, software stack, and training setup.

- In Section E, we summarize dataset statistics for AudioCaps, Clotho, and ESC-50.

- In Section F, we detail evaluation metrics and baselines.

- In Section G, we list the complete hyperparameter settings for all encoder configurations.

- In Section H, we analyze the effect of the weighting parameter $\lambda$ in Eq. equation 14.

- In Section I, we study alternative marginal distributions for the feature-level loss.

- In Table 9, we evaluate the effect of temperature values on retrieval performance.

- In Table 10, we conduct ablation studies on the two loss components $\mathcal{L}_{\text{IOT}}$ and $\mathcal{L}_{\text{UWD}}$.

- In Table 11, we show the effectiveness of $\mathcal{L}_{\text{UWD}}$ as a complementary constraint under different sample-level objectives.

- In Table 12, we investigate the impact of varying mini-batch sizes on DART's performance.

## A  ALGORITHM

---

**Algorithm 1** DART

---

**Input:** Initialize audio encoder $f_\theta$, text encoder $g_\phi$, training data pairs $D$, number of mini-batches $B$

**repeat**

    **for** $b = 1$ **to** $B$ **do**

        Sample $(X^b, Y^b)$ from $D$;

        Embeddings $\mathbf{U}^b = f_\theta(X^b)$, $\mathbf{V}^b = g_\phi(Y^b)$;

        Compute cost matrices $\mathbf{C}^b_{\text{Sample}}$, $\mathbf{C}^b_{\text{Feature}}$ by 4 and 7;

        Solve EOT that $\mathbf{\Pi}^b = \text{EOT}(\mathbf{C}^b_{\text{Sample}})$ by 1;

        $\mathcal{L}_{\text{IOT}} = \text{KL}(\widetilde{\mathbf{\Pi}}^b \| \mathbf{\Pi}^b)$ with label $\widetilde{\mathbf{\Pi}}^b = \text{eye}(\frac{1}{k})$;

        Solve UOT that $\mathbf{P}^b = \text{UOT}(\mathbf{C}^b_{\text{Feature}})$ by 8;

        $\mathcal{L}_{\text{UWD}} = \langle \mathbf{C}^b_{\text{Feature}}, \mathbf{P}^b \rangle$

        $\mathcal{L}_{\text{total}} = \mathcal{L}_{\text{IOT}} + \lambda \mathcal{L}_{\text{UWD}}$

        $\theta, \phi \leftarrow \theta, \phi - \eta \nabla_{\theta,\phi} \mathcal{L}_{\text{total}}$;

    **end for**

**until** converged

---

# B  RELIABILITY SCORE COMPUTATION

In this section, we provide the detailed implementation of the reliability-aware marginal estimation module described in Section 3.3. The procedure consists of three steps: computing channel-wise statistics, aggregating them into reliability scores, and stabilizing the estimates across training.

**Channel-wise Statistics.**  For a mini-batch of size $k$, the audio and text embeddings are denoted as $\mathbf{U}^b, \mathbf{V}^b \in \mathbb{R}^{k \times d}$. For the $j$-th feature channel, we compute:

1. **Cross-modal correlation:**

$$\text{corr}(\mathbf{U}^b(:,j), \mathbf{V}^b(:,j)) = \frac{\langle \mathbf{U}^b(:,j) - \overline{u}_j, \ \mathbf{V}^b(:,j) - \overline{v}_j \rangle}{\|\mathbf{U}^b(:,j) - \overline{u}_j\|_2 \cdot \|\mathbf{V}^b(:,j) - \overline{v}_j\|_2}, \tag{18}$$

   where $\overline{u}_j$ and $\overline{v}_j$ are the sample means of the $j$-th channel. This normalized correlation measures semantic consistency between modalities.

2. **Variance instability:**

$$\text{var}(\mathbf{U}^b(:,j), \mathbf{V}^b(:,j)) = \text{Var}(\mathbf{U}^b(:,j)) + \text{Var}(\mathbf{V}^b(:,j)). \tag{19}$$

   Larger values indicate unstable or noisy activations across samples.

3. **Kurtosis (heavy-tailedness):**

$$\text{kurt}(\mathbf{U}^b(:,j), \mathbf{V}^b(:,j)) = \text{Kurt}(\mathbf{U}^b(:,j)) + \text{Kurt}(\mathbf{V}^b(:,j)), \tag{20}$$

   where $\text{Kurt}(z) = \frac{\mathbb{E}[(z-\mu)^4]}{\sigma^4}$ denotes standardized fourth-order moment. High kurtosis indicates outliers or bursty patterns.

**Score Aggregation.**  The channel reliability score $r_j$ is defined as:

$$r_j = \sigma\Big(\text{corr}(\mathbf{U}^b(:,j), \mathbf{V}^b(:,j)) - \text{var}(\mathbf{U}^b(:,j), \mathbf{V}^b(:,j)) - \text{kurt}(\mathbf{U}^b(:,j), \mathbf{V}^b(:,j))\Big), \tag{21}$$

where $\sigma(\cdot)$ is the sigmoid function. Higher values indicate more stable and semantically reliable channels. The scores are normalized into probability marginals:

$$\boldsymbol{u}^b = \frac{r}{\sum_j r_j}, \quad \boldsymbol{v}^b = \frac{r}{\sum_j r_j}, \tag{22}$$

where $r = (r_1, \ldots, r_d)$.

**Stabilization Across Training.**  Since $r_j$ is computed per mini-batch, it can fluctuate due to randomness in sampling. To stabilize the estimates, we adopt:

- **EMA smoothing:** Reliability scores are updated across training iterations using exponential moving average:
$$r_j^{(t)} \leftarrow \beta \cdot r_j^{(t-1)} + (1 - \beta) \cdot r_j^{\text{batch}},$$
  where $\beta \in [0, 1)$ is a smoothing coefficient.

- **Hysteresis rule:** Thresholds $\tau_{\text{hi}}, \tau_{\text{lo}}$ decide whether to activate or suppress a channel, avoiding oscillations.

- **Warm-up:** All channels are considered active during the first $T_{\text{warm}}$ iterations.

- **Freeze:** After a fixed epoch, the selection of reliable channels can be frozen for stability.

- **Top-$K$ filtering:** Optionally, only the $K$ most reliable channels are retained to reduce noise further.

**Implementation Note.**  All reliability statistics (correlation, variance, kurtosis) are computed independently from the forward-backward graph and can be executed off-GPU. This avoids additional GPU memory usage and ensures minimal runtime overhead.

## C   THEORETICAL ANALYSIS

### C.1   SYMBOL DEFINITIONS

Let the full-batch optimal coupling matrix be $\mathbf{\Pi}^* \in \mathbb{R}^{n \times n}$ for $n$ data pairs. For mini-batch stochastic optimization:

- Let $S_b, T_b \subset [n]$ denote the sample index sets of size $m$ for batch $b$.
- Let $\alpha_b : [m] \to S_b$ and $\beta_b : [m] \to T_b$ be index mapping functions that link local mini-batch indices to global indices.
- Let $\mathbf{\Pi}^b \in \mathbb{R}^{m \times m}$ denote the local coupling matrix for batch $b$, where $\mathbf{\Pi}(p, q)$ corresponding to the pair $(\alpha_b(p), \beta_b(q))$ in the global coupling matrix.

Then we formalize two distinct mini-batch sampling paradigms:

**Definition 3** (Global coupling matrix under Non-overlapping Mini-Batch Sampling). *Let* $\mathbf{\Pi}^B \in \mathbb{R}^{n \times n}$ *be the global coupling matrix constructed from* $B$ *non-overlapping mini-batches* $\{(S_b, T_b)\}_{b=1}^B$:

$$\mathbf{\Pi}^B(i,j) = \begin{cases} \mathbf{\Pi}^b(\alpha_b^{-1}(i), \beta_b^{-1}(j)), & if \quad \exists b \ s.t. \ i \in S_b \ and \ j \in T_b, \\ 0, & otherwise. \end{cases} \tag{23}$$

**Proof Roadmap.**   The theoretical analysis proceeds in three steps:

- **Inner Optimization (Coupling Matrix).** We first establish the $\epsilon$-strong convexity of the entropy-regularized OT objective (Lemma 4), which allows bounding deviations between the mini-batch coupling $\mathbf{\Pi}^B$ and the full-batch optimum $\mathbf{\Pi}^*$ via the functional gap.
- **Mini-Batch Concentration.**   Using Hoeffding's inequality, we derive concentration bounds on the mini-batch OT objective (Lemma 5), showing how the deviation shrinks with the number of batches $B$ and depends on the maximum alignment distance $M$.
- **Loss-Level Bounds.** Finally, we transfer these results to the loss functions: Theorem 1 for the instance-level $\mathcal{L}_{\mathrm{IOT}}$ and Theorem 2 for the feature-level $\mathcal{L}_{\mathrm{UWD}}$, highlighting their different dependence on $D_{\max}$ versus $\|\mathbf{P}^*\|_F$.

Together, these steps show why feature-level regularization yields tighter bounds and better robustness under small batches or noisy data.

### C.2   CONCENTRATION BOUNDS OF $\mathcal{L}_{\mathrm{IOT}}$ (5)

In this subsection, we analyze the deviation introduced by mini-batch optimization in solving the original Inverse Optimal Transport (IOT) problem (5). The IOT problem comprises two sequential stages: (1) an outer minimization stage for learning representations, and (2) an inner optimization stage for computing the coupling (i.e., the probability matching matrix) between point sets. The inner stage corresponds to a standard Entropy-Regularized Optimal Transport (EOT) problem. Our analysis focuses on quantifying the discrepancy between the full-batch solution $\mathbf{\Pi}^*$ and the mini-batch solution $\mathbf{\Pi}^B$.

**Lemma 4** ($\epsilon$-Strongly Convexity of EOT Objective). *Consider the objective function* $f(\cdot)$ : $\mathcal{U}(\mu, \nu) \to \mathbb{R}$ *for the Entropy-Regularized Optimal Transport (EOT) problem, defined as:*

$$f(\mathbf{\Pi}) = \left\langle \mathbf{\Pi}, \mathbf{C} \right\rangle - \epsilon H(\mathbf{\Pi}),$$

*where* $\mathcal{U}(\mu, \nu)$ *denotes the set of coupling matrices with marginals* $\mu$ *and* $\nu$, $\mathbf{C}$ *is the cost function, and* $H(\mathbf{\Pi})$ *is the entropy term. Then* $f$ *is* $\epsilon$-*strongly convex over the relative interior of* $\mathcal{U}(\mu, \nu)$. *Specifically, for any feasible coupling matrix* $\mathbf{\Pi}_1 \in \mathcal{U}(\mu, \nu)$ *and the EOT optimizer* $\mathbf{\Pi}^* := \arg\min_{\mathbf{\Pi} \in \mathcal{U}(\mu, \nu)} f(\mathbf{\Pi})$, *it holds that:*

$$\frac{\epsilon}{2}|\mathbf{\Pi}_1 - \mathbf{\Pi}^*|_F^2 \le f(\mathbf{\Pi}_1) - f(\mathbf{\Pi}^*). \tag{24}$$

*Proof.* For any coupling matrix $\mathbf{\Pi} \in \text{relint}(\mathcal{U}(\mu, \nu))$ (where $\Pi_{ij} > 0$), the Hessian operator of $f$ is given by:

$$\nabla^2 f(\mathbf{\Pi}) = \epsilon \cdot \text{diag} \left( 1/\Pi_{ij} \right)_{i,j},$$

where the diagonal operator acts on the vectorized matrix. For any tangent direction $\mathbf{D} \in T_{\mathcal{U}(\mu,\nu)}$, we have:

$$\langle \mathbf{D}, \nabla^2 f(\mathbf{\Pi}) \mathbf{D} \rangle_F = \epsilon \sum_{i,j} \frac{\mathbf{D}_{ij}^2}{\mathbf{\Pi}_{ij}} \geq \epsilon \|\mathbf{D}\|_F^2,$$

where the inequality follows from $\Pi_{ij} \leq 1$ in the probability simplex. This establishes the $\epsilon$-strong convexity. $\square$

According to 4, the Frobenius norm deviation between $\mathbf{\Pi}^*$ and $\mathbf{\Pi}^B$ is upper-bounded by the functional value difference, which allows us to analyze the convergence through the functional gap that: $|\mathbf{\Pi}^B - \mathbf{\Pi}^*|_F^2 \leq \frac{2}{\epsilon}(f(\mathbf{\Pi}^B) - f(\mathbf{\Pi}^*))$.

**Lemma 5** (Concentration Bound of Mini-Batch EOT Objective). *Let $\delta \in (0, 1)$ and $B \geq 1$, we have a bound between $f(\mathbf{\Pi}^B)$ and $f(\mathbf{\Pi}^*)$ depending on the number of batches $B$ that*

$$\left| f(\mathbf{\Pi}^B) - f(\mathbf{\Pi}^*) \right| \leq M \sqrt{\frac{\log(2/\delta)}{2B}}, \tag{25}$$

*with probability at least $1 - \delta$. Here, $M = D + \epsilon(2 \log_2(m) + 1)$.*

*Proof.* The proof can be found in [Fatras et al. (2019), Lemma 3], and we restate it here for better understanding.

To simplify the problem, we first derive an upper bound for the function value $f(\mathbf{\Pi}^b)$ of any feasible coupling matrix $\mathbf{\Pi}^b \in \mathbb{R}^{m \times m}$ within a mini-batch. For any $X_i \sim \mu$ and $Y_j \sim \nu$, we have $\|X_i - Y_j\| \leq M$, implying $\mathbf{C}_{ij}^b \leq M$ for all $(i, j)$. The Shannon entropy $E(\mathbf{\Pi}^b) = -\sum_{1 \leq i,j \leq m} \mathbf{\Pi}_{ij}^b \log \mathbf{\Pi}_{ij}^b$ satisfies $0 \leq E(\mathbf{\Pi}^b) \leq \log_2(m^2)$ where the maximum entropy occurs when $\mathbf{\Pi}^b$ is uniform. Applying the triangle inequality, we have

$$|f(\mathbf{\Pi}^b)| = \left| \langle \mathbf{\Pi}^b, \mathbf{C}^b \rangle - \epsilon H(\mathbf{\Pi}^b) \right| \leq \left| \sum_{1 \leq i,j \leq m} \mathbf{C}_{ij}^b \mathbf{\Pi}_{ij} \right| + \left| \epsilon \left( -\sum_{1 \leq i,j \leq m} \mathbf{\Pi}_{ij}^b \log \mathbf{\Pi}_{ij}^b + 1 \right) \right|$$

$$\leq D \sum_{1 \leq i,j \leq m} \mathbf{\Pi}_{ij}^b + \epsilon(\log_2(m^2) + 1)$$

$$\leq D + \epsilon(2 \log_2(m) + 1) = M. \tag{26}$$

The mini-batch sampling process can be modeled as a sequence of independent trials over all possible mini-batch configurations. For each trial $b \in [B]$, let $\mathbf{1}_b \in \{0, 1\}$, be a Bernoulli random variable indicating whether a specific mini-batch configuration (among the total $\binom{n}{m}^2$ possible configurations) is selected. Therefore, we then have

$$f(\mathbf{\Pi}^B) - f(\mathbf{\Pi}^*) = \frac{1}{B} \sum_{b=1}^{B} w_b, \tag{27}$$

where $w_b = \left( \mathbf{1}_b - 1/(\binom{n}{m}^2) \right) f(\mathbf{\Pi}^b)$. Here, $\mathbf{1}_b - 1/(\binom{n}{m}^2)$ represents the deviation between the actual selection status of the mini-batch pair $(S_b, T_b)$ in trial $b$ and its expected selection probability. $f(\mathbf{\Pi}^b)$ denotes the objective function value corresponding to the selected mini-batch configuration, and $w_b$ reflects the contribution of trial $b$ to the overall deviation from the expected value. Since the variables $\{w_b\}_{b=1}^B$ are independent, centered ($\mathbb{E}[w_b] = 0$) and bounded by $M$, we can apply Hoeffding's inequality, which gives:

$$\mathbb{P}(|f(\mathbf{\Pi}^B) - f(\mathbf{\Pi}^*)| > t) = \mathbb{P}\left( \left| \frac{1}{B} \sum_{b=1}^{B} w_b \right| > t \right) \leq 2 \exp\left( -\frac{2t^2}{BM^2} \right), \tag{28}$$

To derive a high-probability bound, set the right-hand side equal to a confidence parameter $\delta$: $2\exp(-\frac{2t^2}{BM^2}) = \delta$, solving for $t$ yields:

$$t = M\sqrt{\frac{\log(2/\delta)}{2B}}. \tag{29}$$

$\square$

**Theorem 6** (Maximal Deviation Bound in the Inner Optimization Stage of $\mathcal{L}_{\text{IOT}}$ (5)). *Let $\delta \in (0,1)$, $B \geq 1$ and the mini-batch size $m$ be fixed, we have a maximal deviation bound between $\mathbf{\Pi}^B$ and $\mathbf{\Pi}^*$ that*

$$|\mathbf{\Pi}^B - \mathbf{\Pi}^*|_F^2 \leq \frac{\epsilon}{2}\left|f(\mathbf{\Pi}^B) - f(\mathbf{\Pi}^*)\right|$$

$$\leq \frac{\epsilon M}{2}\left(\sqrt{\frac{\log(2/\delta)}{2B}}\right), \tag{30}$$

*with probability at least $1 - \delta$. Here, $M = D + \epsilon(2\log_2(m) + 1)$.*

*Proof.* Here, the first inequality is obtained by the strong convexity of EOT (24). The second inequality follows from the triangle inequality by introducing the intermediate solution $\mathbf{\Pi}^{\text{exh}}$. Applying 5 to bound the mini-batch estimation error, we derive the final result. $\square$

Before deriving the deviation bound in the outer optimization stage of $\mathcal{L}_{\text{IOT}}$, recall that the mini-batch objective function is defined as

$$\mathcal{L}_{\text{IOT}}^B = -\sum_{b=1}^{B}\sum_{i=1}^{k}\widetilde{\mathbf{\Pi}}_{ii}^b \log(\mathbf{\Pi}_{ii}^{(\theta,\phi)b}), \tag{31}$$

where $\widetilde{\mathbf{\Pi}}$ is a permutation matrix representing the ground-truth matching, with $\widetilde{\mathbf{\Pi}}_{ii} = 1$ indicating correctly paired audio and text instances.

The full-batch loss function is then given by:

$$\mathcal{L}_{\text{IOT}}^* = -\iint_{X \times Y} \tilde{\pi}(x,y)\log_{\pi_{\theta,\phi}}(x,y)\,d\mu(x)d\nu(y). \tag{32}$$

Here, $\mu$ and $\nu$ are discrete measures defined on the finite sets $X = \{x_1, x_2, \ldots, x_m\}$ and $Y = \{y_1, y_2, \ldots, y_n\}$, respectively: $\mu = \frac{1}{m}\sum_{i=1}^{m}\delta_{x_i}$, $\nu = \frac{1}{n}\sum_{j=1}^{n}\delta_{y_j}$. The function $\tilde{\pi}(x,y)$ represents the probabilistic coupling between the elements of $X$ and $Y$.

**Theorem 7** (Concentration Bound of $\mathcal{L}_{\text{IOT}}^B$ (5)). *Let $\delta \in (0,1)$, $B \geq 1$ and the mini-batch size $m$ be fixed. Suppose the log function is $L-$Lipschitz over $[\epsilon, 1]$ for some $\epsilon > 0$, where $\mathbf{\Pi} \in [\epsilon, 1]$ for all batches. Then, with probability at least $1 - \delta$, the maximal deviation between the mini-batch loss $\mathcal{L}_{IOT}^B$ and the full-batch loss $\mathcal{L}_{IOT}^*$ satisfies:*

$$\left|\mathcal{L}_{IOT}^B - \mathcal{L}_{IOT}^*\right|^2 = L^2\left|\mathbf{\Pi}^B - \mathbf{\Pi}^*\right|_F^2 \tag{33}$$

$$\leq \frac{\epsilon M L^2}{2}\left(\sqrt{\frac{\log(2/\delta)}{2B}}\right). \tag{34}$$

*Here, $M = D + \epsilon(2\log_2(m) + 1)$.*

*Proof.* Step 1: Lipschitz continuity of loss difference. Since $\widetilde{\mathbf{\Pi}}$ is a permutation matrix, it satisfies $\tilde{\pi}_{ii} = 1$ for correctly paired instances and 0 otherwise. Thus, the loss function simplifies to:

$$\mathcal{L}_{\text{IOT}}^B = -\frac{1}{B}\sum_{b=1}^{B}\log\mathbf{\Pi}_{ii}^b = -\log\mathbf{\Pi}_{ii}^B, \quad \mathcal{L}_{\text{IOT}}^* = -\log\mathbf{\Pi}_{ii}^*. \tag{35}$$

Both $\mathbf{\Pi}_{ii}^B$ and $\mathbf{\Pi}^*$ correspond to the predicted probabilities of correct matches, their values tend to be close to 1. Therefore, assuming that the logarithm function is $L-$ Lipschitz over the domain of $\mathbf{\Pi}_{ii}$ is reasonable. This implies the following bound:

$$\left|\mathcal{L}_{\text{IOT}}^B - \mathcal{L}_{\text{IOT}}^*\right| = \left|\log \mathbf{\Pi}_{ii}^* - \log \mathbf{\Pi}_{ii}^B\right| \tag{36}$$

$$\leq L \cdot \left|\mathbf{\Pi}_{ii}^* - \mathbf{\Pi}_{ii}^B\right| \quad \text{(Lipschitz condition)} \tag{37}$$

$$\leq L \cdot \sqrt{\sum_{i=1}^m (\mathbf{\Pi}_{ii}^* - \mathbf{\Pi}_{ii}^B)^2} \quad \text{(Element-wise difference to vector 2-norm)} \tag{38}$$

$$\leq L \cdot \|\mathbf{\Pi}^* - \mathbf{\Pi}^B\|_F \quad \text{(Vector 2-norm to matrix Frobenius norm)}. \tag{39}$$

Step 2: Concentration via pre-established bound. By applying the concentration result in 6, we directly obtain the 33. $\qquad\square$

## C.3 CONCENTRATION BOUND OF $\mathcal{L}_{\text{UWD}}$ (9)

**Assumption 8** (Statistical Model of Feature Extractors). *The feature encoders $f_\theta$ and $g_\phi$ satisfy:*

$$f_\theta(x)_i = \mu_{f,i} + \delta_{f,i}(x), \quad \mathbb{E}_{x\sim\mathcal{X}}[\delta_{f,i}(x)] = 0, \quad Var_{x\sim\mathcal{X}}(\delta_{f,i}(x)) = \sigma_{f,i}^2, \tag{40}$$

$$g_\phi(y)_j = \mu_{g,j} + \delta_{g,j}(y), \quad \mathbb{E}_{y\sim\mathcal{Y}}[\delta_{g,j}(y)] = 0, \quad Var_{y\sim\mathcal{Y}}(\delta_{g,j}(y)) = \sigma_{g,j}^2, \tag{41}$$

*where $\mu_{f,i}, \mu_{g,j}$ are expected feature values, and $\delta_{f,i}(x), \delta_{g,j}(y)$ are zero-mean noise terms.*

**Lemma 9** (Concentration of Feature-Level Cost Matrix). *Let $Z_k := f_\theta(x_k)_i - g_\phi(y_k)_j$ denote the difference in the $i$-th and $j$-th feature dimensions of paired embeddings, where $\{(x_k, y_k)\}_{k=1}^m$ are sampled i.i.d., and suppose each $Z_k$ is sub-Gaussian with zero mean and bounded variance proxy $\sigma^2$ (i.e., $\mathbb{E}[Z_k] = 0$, $\mathbb{E}[Z_k^2] \leq \sigma^2$). Given the empirical cost entry as:*

$$\mathbf{C}_{ij}^{(Feat)B} := \left\|f^{(i)} - g^{(j)}\right\|_2 = \sqrt{\sum_{k=1}^m Z_k^2},$$

*and let the full-batch cost be: $\mathbf{C}_{ij}^{(Feat)*} := \sqrt{m} \cdot \left|\mathbb{E}[f_\theta(x)_i] - \mathbb{E}[g_\phi(y)_j]\right|$. Then, with probability at least $1 - \delta$, the deviation between empirical and expected feature-level cost satisfies:*

$$\left|\mathbf{C}_{ij}^{(Feat)B} - \mathbf{C}_{ij}^{(Feat)*}\right| \leq \sqrt{\frac{2\sigma^2 \log(2/\delta)}{m}}.$$

*Proof.* We begin by computing the expectation of $\mathbf{C}_{ij}^{(\text{Feat})B}$:

$$\mathbb{E}\left[\mathbf{C}_{ij}^{(\text{Feat})B}\right] = \mathbb{E}\left[\sum_{k=1}^m (f_\theta(x_k)_i - g_\phi(y_k)_j)^2\right] = \sum_{k=1}^m \mathbb{E}\left[((\mu_{f,i} + \delta_{f,k}) - (\mu_{g,j} + \delta_{g,k}))^2\right]$$

$$= \sum_{k=1}^m \mathbb{E}\left[((\mu_{f,i} - \mu_{g,j}) + (\delta_{f,k} - \delta_{g,k}))^2\right]$$

$$= \sum_{k=1}^m \mathbb{E}\left[(D_{ij} + \delta_k)^2\right] = \sum_{k=1}^m (D_{ij}^2 + 2D_{ij}\mathbb{E}[\delta_k] + \mathbb{E}[\delta_k^2])$$

$$= m\left(\sigma^2 + D_{ij}^2\right). \tag{42}$$

To simply, we set $C_{ij}^{(\text{Feat})B} = \sqrt{\sum_{k=1}^m Z_k^2}$, where $Z_k := f_\theta(x_k)_i - g_\phi(y_k)_j$. Since $Z_k$ are sub-Gaussian, and the squared terms $Z_k^2$ are sub-exponential. According to the concentration inequality for sub-exponential variables, we have:

$$\mathbb{P}\left(\left|\frac{1}{m}\sum_{k=1}^m Z_k^2 - \mathbb{E}[Z_k^2]\right| \geq t\right) \leq 2\exp\left(-c \cdot m \cdot \min\left(\frac{t^2}{\sigma^4}, \frac{t}{\sigma^2}\right)\right). \tag{43}$$

According 42, we substitute $\mathbb{E}[Z_k^2] = D_{ij}^2 + \sigma^2$ and $S = \sum_{k=1}^m Z_k^2$, for $t = \epsilon\sqrt{m}$ and obtain:

$$\mathbb{P}\left(\left|S - m(D_{ij}^2 + \sigma^2)\right| \geq \epsilon m\right) \leq 2\exp\left(-\frac{c\epsilon^2 m}{\sigma^4}\right). \tag{44}$$

Using the inequality

$$\left|\sqrt{S} - \sqrt{\mathbb{E}[S]}\right| \leq \frac{|S - \mathbb{E}[S]|}{2\sqrt{\mathbb{E}[S]}}, \quad \text{with } \mathbb{E}[S] = m(D_{ij}^2 + \sigma^2),$$

we derive:

$$\mathbb{P}\left(\left|\mathbf{C}_{ij}^{(\text{Feat})B} - \sqrt{m(D_{ij}^2 + \sigma^2)}\right| \geq \epsilon\right) \leq \mathbb{P}\left(\left|S - m(D_{ij}^2 + \sigma^2)\right| \geq 2\epsilon\sqrt{m(D_{ij}^2 + \sigma^2)}\right). \tag{45}$$

Combining with the result, we obtain:

$$\mathbb{P}\left(\left|\mathbf{C}_{ij}^{(\text{Feat})B} - \sqrt{m(D_{ij}^2 + \sigma^2)}\right| \geq \epsilon\right) \leq 2\exp\left(-\frac{2\epsilon^2 m(D_{ij}^2 + \sigma^2)}{\sigma^4}\right). \tag{46}$$

Recall that $\mathbf{C}_{ij}^{(\text{Feat})*} = \sqrt{m}D_{ij}$, and that $\sqrt{m(D_{ij}^2 + \sigma^2)} \leq \sqrt{m}D_{ij} + \frac{\sigma^2}{2D_{ij}}$ (via Taylor expansion), we can write:

$$\left|\mathbf{C}_{ij}^{(\text{Feat})B} - \mathbf{C}_{ij}^{(\text{Feat})*}\right| \approx \left|\sqrt{S} - \sqrt{m}D_{ij}\right| \leq \epsilon + \frac{\sigma^2}{2D_{ij}}. \tag{47}$$

$\square$

**Theorem 10** (Feature-level Loss $\mathcal{L}_{\text{UWD}}$ (9) Concentration). *Assume the feature-level cost matrix $\mathbf{C}_{ij}^{(\text{Feat})B} \in \mathbb{R}^{d \times d}$ is computed from $m$ i.i.d. paired samples, and each entry satisfies the deviation bound:*

$$\left|\mathbf{C}_{ij}^{(\text{Feat})B} - \mathbf{C}_{ij}^{(\text{Feat})*}\right| \leq \epsilon_m \quad \textit{with probability at least } 1 - \delta,$$

*where $\epsilon_m = \sqrt{\frac{2\sigma^2 \log(2/\delta)}{m}}$. Then the unbalanced OT loss satisfies the deviation bound:*

$$\left|\mathcal{L}_{UOT}(\mathbf{C}_{ij}^{(\text{Feat})B}) - \mathcal{L}_{UOT}(\mathbf{C}_{ij}^{(\text{Feat})*})\right| \leq \|\mathbf{P}^*\|_F \cdot \epsilon_m + \frac{1}{2\lambda} \cdot \epsilon_m^2,$$

*with probability at least $1 - \delta$, where $\lambda = \epsilon + reg_m$ is the strong convexity constant of the UOT objective.*

*Proof.* Leveraging the strong convexity of unbalanced optimal transport (with coefficient $\lambda = \epsilon + reg_m$):

$$\left|\mathcal{L}_{\text{UOT}}(\mathbf{C}_{ij}^{(\text{Feat})B}) - \mathcal{L}_{\text{UOT}}(\mathbf{C}_{ij}^{(\text{Feat})*})\right| \leq \|\mathbf{P}^*\|_F \cdot \|\mathbf{C}^{(\text{Feat})B} - \mathbf{C}^{(\text{Feat})*}\|_F + \frac{1}{2\lambda}\|\mathbf{C}^{(\text{Feat})B} - \mathbf{C}^{(\text{Feat})*}\|_F^2. \tag{48}$$

Based on Lemma 9, the theorem follows. $\square$

## D  IMPLEMENTATION DETAILS

The experiments are conducted on a Linux workstation equipped with an Intel(R) Xeon(R) Gold 6226R CPU (2.90GHz) and an NVIDIA A100-PCIE-40GB GPU. The detailed implementation code is provided in the supplementary materials.

# E  DATASETS DETAILS

We evaluate DART on three widely used datasets: **AudioCaps** Kim et al. (2019), **Clotho** Drossos et al. (2020), and **ESC-50** Piczak (2015), covering audio-text retrieval and sound event detection tasks.

**AudioCaps** is the largest audio captioning dataset, containing approximately 50K audio-caption pairs. All audio clips are sourced from AudioSet Gemmeke et al. (2017), a large-scale dataset for audio tagging. The training set consists of 40,582 audio clips, each 10 seconds long and paired with a single human-annotated caption. In contrast, the validation and test sets contain 494 and 957 audio clips, respectively, with each clip accompanied by five ground-truth captions.

**Clotho** is an audio captioning dataset collected from the Freesound platform, featuring audio clips of varying durations between 15 to 30 seconds. We use the second version of the dataset for our experiments. The training set includes 3,839 audio clips, while the validation and test sets contain over 1k clips each. Every audio clip is paired with five human-annotated captions.

**ESC-50** is an environmental sound classification dataset designed for sound event detection, consisting of 2K labelled recordings across 50 sound classes. Since our goal is to evaluate DART's transferability, we use only the test set, which contains 400 audio clips.

**Baselines.**  We compare DART against state-of-the-art audio-text retrieval models, including Oncescu et al. (2021), Mei et al. (2022), Deshmukh et al. (2022), Wu et al. (2023), Luong et al. (2024), Wang et al. (2023) and Chen et al. (2023), ensuring consistency in evaluation settings. All baseline results are directly sourced from their respective papers for fair comparison. Furthermore, we investigate the impact of different training objectives, including contrastive and triplet loss, to analyze DART's adaptability under various learning paradigms.

# F  EXPERIMENTAL SETUP

**Evaluation metrics.**  We evaluate DART using Recall at Rank $k$ (R@$k$), a standard metric for cross-modal retrieval. R@$k$ measures the proportion of queries where at least one ground-truth match appears in the top-$k$ retrieved results. Formally, for a query set of size $N$, R@$k$ is computed as:

$$\text{R@}k = \frac{1}{N} \sum_{i=1}^{N} \mathbb{I}\left(\text{rank}(y_i) \leq k\right), \tag{49}$$

where $\mathbb{I}(\cdot)$ is the indicator function that returns 1 if the correct match $y_i$ is ranked within the top-$k$, and 0 otherwise. A higher R@$k$ indicates better retrieval performance. We report R@1, R@5, and R@10 to compare DART with baseline methods.

# G  HYPERPARAMETERS

Tab. 6 provides a comprehensive overview of the hyperparameters employed across all baseline models. The three sections correspond to different encoder configurations evaluated in the main comparison table: (1) ResNet38 (audio) + BERT (text), (2) CNN (audio) + BPE (text), and (3) Beats (audio) + BERT (text). All settings align with those used in the respective baselines to ensure fair comparison and match the configurations reported in their original papers.

# H  EFFECT OF THE WEIGHTING PARAMETER $\lambda$

We analyze the effect of the weighting parameter $\lambda$ in the overall loss function 14. 7 presents the retrieval performance under different values of $\lambda$. The results indicate that DART is robust to variations in $\lambda$, with consistent performance across the tested range. The best performance is observed at $\lambda = 0.7$, achieving an R@1 score of 40.41% for text-to-audio retrieval and 53.70% for audio-to-text retrieval. Notably, even at $\lambda = 0.1$, the model performs well, with R@1 scores of 40.31% and 53.29% for text-to-audio and audio-to-text retrieval, respectively. This suggests

Table 6: Detailed hyper-parameters used in training for the retrieval experiments reported in Table 1.

| Hyperparameters | AudioCaps | Clotho |
|---|---|---|
| Batch size | 256 | 256 |
| Optimizer | Adam | Adam |
| Learning rate | $5 \times 10^{-5}$ | $5 \times 10^{-5}$ |
| Weight decay | 0.0 | 0.0 |
| Total epoch | 10 | 10 |
| $\lambda$ | 0.5 | 0.5 |
| $\epsilon$ | 0.03 | 0.03 |
| $\tau$ | 0.05 | 0.05 |
| Batch size | 6 | 6 |
| Optimizer | AdamW | AdamW |
| Adam $\beta$ | (0.9,0.999) | (0.9,0.999) |
| Learning rate | $1 \times 10^{-6}$ | $1 \times 10^{-6}$ |
| Weight decay | 0.01 | 0.01 |
| Total epoch | 10 | 10 |
| $\lambda$ | 0.5 | 0.5 |
| $\epsilon$ | 0.03 | 0.03 |
| $\tau$ | 0.05 | 0.05 |
| Batch size | 256 | 256 |
| Optimizer | AdamW | AdamW |
| Adam $\beta$ | (0.9, 0.98) | (0.9, 0.98) |
| Learning rate | $5 \times 10^{-7}$ | $5 \times 10^{-7}$ |
| Weight decay | 0.01 | 0.01 |
| Total epoch | 10 | 10 |
| $\lambda$ | 0.5 | 0.5 |
| $\epsilon$ | 0.03 | 0.03 |
| $\tau$ | 0.05 | 0.05 |

that while the feature-level alignment provided by $\mathcal{L}_{\text{UWD}}$ contributes to optimal performance, the underlying IOT framework also plays a critical role in ensuring DART's robustness. This flexibility underscores DART's ability to effectively balance the contributions of different loss components, enabling robust cross-modal retrieval across diverse settings.

Table 7: Retrieval performance (%) under varying $\lambda$ values (Eq. 14) with a small fixed mini-batch size of 32.

| $\lambda$ | Text $\rightarrow$ Audio | | | Audio $\rightarrow$ Text | | |
|---|---|---|---|---|---|---|
| | R@1 | R@5 | R@10 | R@1 | R@5 | R@10 |
| 0.1 | 40.31 | **75.23** | **86.22** | 53.29 | 83.07 | 90.17 |
| 0.3 | 39.97 | 75.04 | 85.66 | 51.51 | 82.23 | 90.28 |
| 0.5 | 39.94 | 75.07 | 85.64 | 53.08 | **83.49** | **90.59** |
| 0.7 | **40.41** | 75.06 | **86.22** | **53.70** | 81.50 | 90.28 |

## I    EFFECT OF THE MARGINALS IN $\mathcal{L}_{\text{UWD}}$

We analyze the effect of the marginal distributions used in the feature-level loss defined in Eq. 9. Specifically, we consider the following initialization strategies for the source and target marginals:

- **Uniform Distribution**: This baseline assumes no prior knowledge of feature importance and sets both marginals to uniform weights, assigning equal mass to all feature dimensions.
- **Feature Norm-Based Initialization**: In this setting, the marginal distributions are derived from the $\ell_2$ norm of the corresponding feature dimensions. The underlying motivation is that dimensions with higher magnitude may carry more semantic or discriminative information, and thus should be assigned greater mass in the transport plan.
- **Feature Variance-Based Initialization**: Here, we use the empirical variance of each feature dimension across the batch to form the marginals. The rationale is that dimensions exhibiting higher variance across samples are likely to be more informative and discriminative for downstream alignment.

Table 8: Retrieval performance (%) under different marginal distribution in 9.

| Marginal | Text $\rightarrow$ Audio | | | Audio $\rightarrow$ Text | | |
|---|---|---|---|---|---|---|
| | R@1 | R@5 | R@10 | R@1 | R@5 | R@10 |
| Uniform ($\mathcal{U}$) | 32.87 | 67.77 | 81.06 | 43.57 | **73.98** | **86.72** |
| $L_2$ Norm-based | **33.24** | **68.54** | **81.40** | 42.00 | 73.87 | 85.37 |
| Variance-based | 33.10 | 68.25 | 80.64 | **43.88** | 73.24 | 85.89 |

## J    EFFECT OF THE TEMPERATURE VALUES IN $\mathcal{L}_{\text{UWD}}$

Table 9: Retrieval performance (%) under different temperature values with a small fixed mini-batch size of 32.

| Temperature | Text $\rightarrow$ Audio | | | Audio $\rightarrow$ Text | | |
|---|---|---|---|---|---|---|
| | R@1 | R@5 | R@10 | R@1 | R@5 | R@10 |
| 0.05 | 36.46 | 71.95 | 83.94 | 46.39 | 78.05 | 88.29 |
| 1.0 | 35.59 | 71.43 | 83.97 | 48.69 | 77.33 | 86.42 |
| 10.0 | 37.47 | 71.60 | 83.72 | 46.81 | 75.76 | 86.00 |
| 100.0 | 35.34 | 71.37 | 83.85 | 46.71 | 78.89 | 89.13 |
| inf (eot) | 36.13 | 72.20 | 83.66 | 46.81 | 77.74 | 88.09 |

# K ABLATION STUDY ON THE TWO LOSS

We analyze the contribution of individual loss components to the overall performance. 12 presents the results of the ablation study on the loss components, comparing the effects of $\mathcal{L}_{\text{IOT}}$ and $\mathcal{L}_{\text{UWD}}$ individually and in combination. Using only $\mathcal{L}_{\text{UWD}}$ results in poor performance, with R@1 scores close to zero, highlighting the necessity of correspondence labels for cross-modal alignment tasks. When both $\mathcal{L}_{\text{IOT}}$ and $\mathcal{L}_{\text{UWD}}$ are combined, the model achieves the best performance, which demonstrates the complementary nature of the two loss components.

Table 10: Ablation study on the training loss components using the AudioCaps dataset with a small fixed mini-batch size of 128.

| LOSS | Text → Audio | | | Audio → Text | | |
|---|---|---|---|---|---|---|
| | R@1 | R@5 | R@10 | R@1 | R@5 | R@10 |
| $\mathcal{L}_{\text{UWD}}$ | 0.12 | 0.52 | 0.98 | 0.10 | 0.31 | 0.41 |
| $\mathcal{L}_{\text{IOT}}$ | 39.28 | 74.50 | 85.85 | 52.45 | 80.25 | 90.17 |
| $\mathcal{L}_{\text{IOT}} + \mathcal{L}_{\text{UWD}}$ | **39.94** | **75.07** | **85.64** | **53.08** | **83.49** | **90.59** |

# L EFFECTIVENESS OF FEATURE-LEVEL LOSS AS A COMPLEMENTARY CONSTRAINT

Our feature-level loss $\mathcal{L}_{\text{UWD}}$ is designed to capture fine-grained alignment between modality-specific dimensions, and is intended to be used as a complementary constraint rather than a standalone objective. Specifically, while many existing retrieval systems are trained with sample-level objectives such as contrastive loss, triplet loss, or more advanced models like m-LTM, our method is compatible with all of them.

In this section, we conduct a controlled ablation study to demonstrate that $\mathcal{L}_{\text{UWD}}$ can be seamlessly integrated with different sample-level losses and consistently improves retrieval performance across the board. Table 11 summarizes the results on the AudioCaps dataset (same setup as in the main paper). We observe that for each baseline loss, adding our feature-level loss leads to noticeable gains in both A→T and T→A retrieval tasks.

This confirms that our approach serves as a modular and universally beneficial component that strengthens the representation alignment across modalities without conflicting with the core retrieval objective.

Table 11: Retrieval performance on the AudioCaps dataset with different sample-level objectives, evaluated with and without the feature-level loss $\mathcal{L}_{\text{UWD}}$ under a small fixed mini-batch size of 32.

| Method | A→T R@1 | T→A R@1 |
|---|---|---|
| Triplet loss | 37.72 | **32.85** |
| + $\mathcal{L}_{\text{UWD}}$ | **38.24** | 32.10 |
| Contrastive loss | 38.24 | 31.07 |
| + $\mathcal{L}_{\text{UWD}}$ | **39.18** | **32.14** |
| IOT loss Luong et al. (2024) | 41.69 | 32.39 |
| + $\mathcal{L}_{\text{UWD}}$ | **41.79** | **32.87** |

## L.1 EFFECT OF BATCH SIZES (NUMBER OF SAMPLES PER BATCH)

Second, we examine the impact of batch size on DART's performance, particularly focusing on the role of $\mathcal{L}_{\text{UWD}}$. As shown in 12, the benefits of $\mathcal{L}_{\text{UWD}}$ are more pronounced with smaller batch sizes. This finding is particularly relevant for real-world applications where computational resources are often constrained. By providing feature-level alignment, $\mathcal{L}_{\text{UWD}}$ enables DART to maintain strong

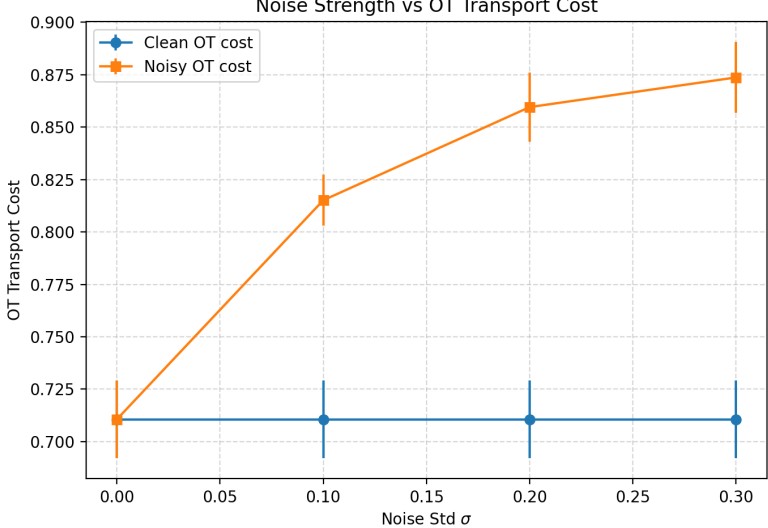

Figure 2: Effect of injected noise on OT cost. We fix a clean pair of audio–text feature distributions and inject i.i.d. Gaussian noise with increasing standard deviation $\sigma$ into one modality. For each $\sigma$, we plot the OT cost between clean features (clean→clean) and between noisy and clean features (noisy→clean). The noisy OT cost grows monotonically with $\sigma$ and is consistently larger than the clean baseline, empirically supporting the claim that stronger channel noise leads to larger transport costs.

performance despite having fewer negative samples, making it well-suited for large-scale deployments with limited resources.

Table 12: Retrieval Performance (%) with Varying Mini-Batch Sizes (Number of Samples per Batch).

| $k$ | LOSS | Text → Audio | | | Audio → Text | | |
|---|---|---|---|---|---|---|---|
| | | R@1 | R@5 | R@10 | R@1 | R@5 | R@10 |
| 8 | SOTA Luong et al. (2024) | 20.44 | 49.95 | 65.54 | 32.91 | 63.74 | 77.11 |
| | $\mathcal{L}_{\text{IOT}} + \mathcal{L}_{\text{UWD}}$ | 24.24 | 57.57 | 72.49 | 35.21 | 65.93 | 78.78 |
| 32 | SOTA Luong et al. (2024) | 33.77 | 69.94 | 82.44 | 43.36 | 74.19 | 85.78 |
| | $\mathcal{L}_{\text{IOT}} + \mathcal{L}_{\text{UWD}}$ | 36.46 | 71.95 | 83.94 | 46.39 | 78.05 | 88.29 |
| 128 | SOTA Luong et al. (2024) | 39.28 | 74.50 | 85.85 | 52.45 | 80.25 | 90.17 |
| | $\mathcal{L}_{\text{IOT}} + \mathcal{L}_{\text{UWD}}$ | 39.94 | **75.07** | **85.64** | **53.08** | **83.49** | **90.59** |

# M ABLATION STUDY FOR RAM

Table 13: Corr-based RAM variants on AudioCaps (ResNet38–BERT, batch size 64).

| Marginal Design | A→T R@1 | T→A R@1 | Mean R@1 |
|---|---|---|---|
| corr-gap | 51.83 | 38.12 | 44.97 |
| corr-burt | 50.99 | 38.75 | 44.87 |

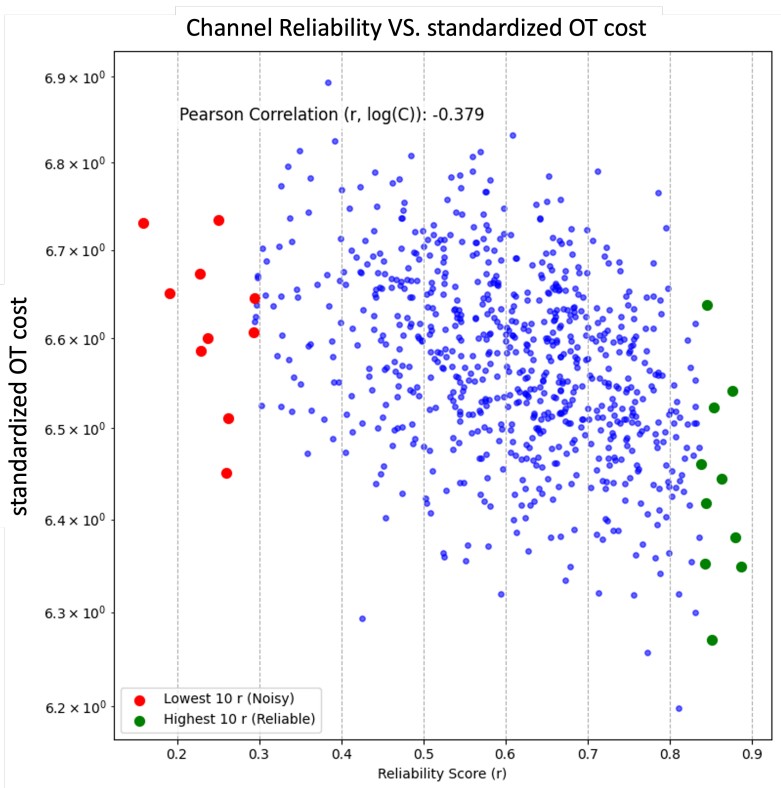

Figure 3: Relationship between reliability scores and standardized OT cost in a trained DART model. For each feature channel $j$, we compute its reliability score $r_j$ and standardized OT cost $\tilde{C}_j$. Each point corresponds to one channel, and we report the Pearson correlation $\rho(r, \log \tilde{C})$ in the legend. Channels with low reliability (highlighted in red) concentrate in the high-cost region, while high-reliability channels (highlighted in green) lie in the low-cost region, indicating that RAM successfully down-weights noisy, high-cost channels in the transport.

## N  VISUALIZATION OF THE PROPOSED RAM

To verify that RAM truly identifies and suppresses noisy channels, we relate our reliability scores $\{r_j\}$ to the *transport signal* produced by DART itself. In a trained model, the feature-level UWD module induces a channel-wise transport cost reflecting the inherent difficulty of aligning each channel across modalities. We denote this empirical, solver-induced quantity as the *actual OT cost*. Since raw costs vary with feature scale and batch statistics, we standardize them to obtain $\tilde{C}_j$, removing scale and batch effects to ensure comparability across channels. A larger $\tilde{C}_j$ means channel $j$ is less consistent cross-modally (more noise-like or modality-specific), and thus should receive less transport mass.

As shown in Fig. 3, $r_j$ is negatively correlated with the standardized OT cost, with Pearson $\rho(r, \log \tilde{C}) \approx -0.379$. Channels with the lowest reliability scores (red) concentrate in the high-cost region, while the highest-reliability channels (green) lie predominantly in the low-cost region. This result supports two key points. First, the transport objective itself assigns higher costs to channels that behave noisily or inconsistently across modalities, corroborating our intuition that such channels are poor carriers of shared semantics. Second, RAM is well aligned with this transport cost structure: by using $r_j$ to form reliability-aware marginals, RAM assigns lower marginal weights to high-cost channels and biases the transport plan toward low-cost channels that more consistently encode stable semantic cues. Consequently, spurious alignment signals from a small set of volatile channels are less likely to dominate the matching process, and transport mass is concentrated on channels that support robust cross-modal alignment.

