# OpenReview forum: "Beyond Instance-Level Alignment: Dual-Level Optimal Transport for Audio-Text Retrieval"
_ICLR.cc/2026/Conference — ICLR 2026 Poster_

### Official Review · Reviewer_BWQA · 2025-10-31

**Soundness:** 2
**Presentation:** 1
**Contribution:** 2
**Rating:** 4
**Confidence:** 3

**Summary:**

The paper proposes DART, a framework for cross-modal retrieval that addresses the limitations of purely instance-level alignment under small batch sizes and scarce labels. DART combines the conventional instance-level IOT objective with a novel feature-level regularization based on the UWD. It employs RAM to reweight feature channels based on statistical cues (correlation, variance, kurtosis), guiding the transport plan towards stable semantic dimensions and suppressing noisy ones. The framework demonstrates state-of-the-art results on audio-text benchmarks, and generalizing to image-text retrieval.

**Strengths:**

1. The introduced feature-level alignment using UWD is new. It tries to address the known weakness of traditional contrastive losses (and instance-level IOT) that treat all feature dimensions equally.
2. DART achieves better performance compared to previous works.
3. Experiments on image-text retrieval shows the generalization ability of the proposed DART.

**Weaknesses:**

1. The writing of this paper is below ICLR's standard. For example, in Line 137 and 142, ``??'' apprear in the main text instead of the reference to equations, figures, or tables. In Eq. (6), x and y lack explanations.
2. Sec. 2.3, limitations of instance-level IOT lacks of theoretical or experimental evidence to support the claims.
3. Novelty is limited: The idea of learning feature importance (reweighting dimensions) is not new in cross-modal retrieval, as noted by the authors themselves when discussing Luong et al. (2024) . While DART's method uses cross-channel transport and richer statistics, the incremental novelty over existing per-channel weighting schemes is largely due to the expensive UWD machinery, which leads to a major concern about complexity.
4. RAM is based on simple, first-order statistics (cross-modal correlation, variance, and kurt). These static, hand-designed proxies for ``semantic stability'' may be insufficient for highly complex, evolving feature spaces. The reliance on simple statistics makes the mechanism feel more like a heuristic than a deep, learned principle. The ablation also shows that RAM improves performance (Tab 1, DART w/ RAM vs w/o RAM) with limited improvements.
5. Also, it would be interesting to see which part of RAM is most important.
6. Lack of running time, memory comparison.

**Questions:**

1. See weakness
2. In eq. 10, how to calculate kurt?

---

> ### Author Response · Authors · 2025-12-03
>
> **Q1**: The writing of this paper is below ICLR's standard. For example, in Line 137 and 142, “??” appears in the main text instead of the reference to equations, figures, or tables. In Eq. (6), x and y lack explanations.
>
> **R1**: Thank you for pointing this out. We have carefully revised the paper for clarity and correctness.
>
> **Q2**: Sec. 2.3, limitations of instance-level IOT lacks theoretical or experimental evidence to support the claims.
>
> **R2** The limitation described in Sec. 2.3 was already supported by our theory in Sec. 4; in the revision we now make this connection explicit and complement it with additional empirical evidence. Thank you for pointing this out.
>
> * Theoretical Evidence (Section 4): In Sec. 2.3 we argue that mini-batch instance-level IOT can be dominated by a few noisy channels, because all embedding dimensions are aggregated uniformly when computing the instance distance $d(x_i, y_j)$. This intuition is made precise in Theorem 1 in Sec. 4. There, we derive a concentration bound for the instance-level IOT loss whose deviation is controlled by the maximum alignment distance $D_{\max}$.
> In other words, the concentration bound of the instance-level objective is governed by the worst-case pair in a mini-batch. Noisy or high-variance channels inflate $d(x_i, y_j)$ for some matched pairs, which in turn enlarges $D_{\max}$ and makes the loss highly sensitive to outliers.
>
> * Empirical Evidence (Appendix N): We have added new experiments that directly confirm this issue:Fig. 2 shows that injecting synthetic noise into features systematically leads to a higher Optimal Transport cost, supporting our claim that noisy channels incur large transport costs.Fig. 3 shows that on a trained DART model, our reliability score exhibits a negative correlation with the standardized channel OT cost, confirming that our method actively targets and suppresses the high-cost (noisy) channels in practice.
>
> **Q3**: Novelty is limited; learning feature importance (reweighting dimensions) is not new (e.g. Luong et al. 2024), and DART seems to only combine instance- and feature-level alignment with expensive UWD.
>
> **R3**: Here we clarify how DART goes beyond prior channel-level methods:
>
> First, DART is a different problem formulation, not just combine “instance- and feature-level”. Prior channel-weighting methods operate strictly at the instance level: per-channel coefficients only change how each dimension contributes to a single instance distance $d(x_i, y_j)$ (Euclidean / cosine), and the loss remains an instance-level objective. As shown in Theorem 1, such instance-level objectives have concentration bounds controlled by the worst-case distance $D_{\max}$ in a mini-batch, which makes them sensitive to outliers and noisy channels. Changing per-dimension weights does not alter this structural dependence.
>
> DART, by contrast, steps outside the instance-distance framework by introducing a separate feature-level OT objective. At this level, we align the distributions of feature channels across the batch via an unbalanced Wasserstein distance. Theorem 2 shows that the deviation of this feature-level loss is governed by an aggregate quantity (the Frobenius norm) rather than a single extremal distance, which explains the improved robustness under small batches and noisy labels.
>
> Second, The proposed Reliability-Aware Marginals (RAM) also differ conceptually from diagonal reweighting. RAM acts inside the OT problem: the corr–var–kurt scores are normalized into marginal distributions that control how much transport mass each channel can send or receive. This directly limits the participation of unreliable channels in the global transport and allows reliable semantic channels to carry most of the mass, while supporting flexible cross-channel couplings $(j:k)$ via the transport plan $\mathbf{P}_{\text{Feature}}$. Diagonal scaling methods cannot express such cross-channel assignments.
>
> We believe these two aspects—changing the structure of the objective and introducing a mass-modulation mechanism inside OT—go beyond simply “reweighting dimensions”.

---

> ### Author Response · Authors · 2025-12-03
>
> **Q4**: RAM uses simple statistics (corr, var, kurt) and seems heuristic; the improvement over w/o RAM in Table 1 is limited. Also, which part of RAM is most important?
>
> **R4**: We appreciate this concern and agree that the role of RAM should be clarified. RAM is indeed built from simple statistics (correlation, variance, kurtosis), but this choice is deliberate rather than ad-hoc. These quantities are cheap to compute per mini-batch and easy to update during training, which keeps the method scalable. In contrast, adding a separate deep module to predict reliability would incur much higher computation and memory cost and make the mechanism harder to interpret.
>
> As summarized in Tables R1–R2, we now evaluate the following core variants on AudioCaps (ResNet38–BERT, batch size 64):
>
> - uniform: all channels treated equally (no RAM),
> - corr: correlation only,
> - emavar: EMA variance only,
> - kurt: kurtosis only,
> - RAM (full): corr + EMA variance + kurtosis.
>
> Table R1 (reproduced here for convenience) reports R@1 and R@10 for both directions and the mean R@1:
>
> | Marginal Design        | A→T R@1 | A→T R@10 | T→A R@1 | T→A R@10 | Mean R@1 |
> |------------------------|--------:|---------:|--------:|---------:|---------:|
> | uniform (w/o RAM)      |   51.52 |    90.80 |   38.31 |    85.77 |    44.92 |
> | corr (correlation)     |   50.05 |    90.60 |   38.64 |    85.22 |    44.35 |
> | emavar (EMA variance)  |   51.83 |    90.49 |   38.52 |    85.56 |    45.18 |
> | kurt (kurtosis)        |   51.93 |    90.60 |   38.64 |    85.74 |    45.29 |
> | RAM (full)             |   52.56 |    90.60 |   38.54 |    85.56 |    45.55 |
>
> Our ablations also show that these statistics are necessary in practice. The corr-only variant underperforms the uniform baseline in mean R@1, confirming that correlation alone is unstable and can be dominated by spurious signals. In contrast, the variance-only (emavar) and kurtosis-only (kurt) variants both improve over the baseline, indicating that they each act as effective stabilizers. Most importantly, the full RAM combination (corr–var–kurt) achieves the best overall performance. This suggests that combining correlation, variance, and kurtosis is essential to obtain both stability and peak accuracy, and that RAM goes beyond a purely ad-hoc heuristic.
>
> Regarding the “limited improvement” over w/o RAM, our main contribution lies in the DART framework itself, which introduces a new dual-level OT formulation compatible with existing instance-level methods. RAM is deliberately designed as a light-weight plug-in on top of this strong baseline: it adds negligible cost, is easy to implement with any encoder, and still delivers consistent gains. The goal is not to replace DART’s core alignment mechanism, but to turn the noisy-channel intuition into a small yet measurable robustness improvement across different settings.
>
>
> **Q5**: Lack of running time and memory comparison; complexity concerns about UWD.
>
> **R5**: We have added a runtime and memory comparison in Appendix J to address this concern.
>
> Memory. In our main experiments, the encoder dimensionality is at most \(d = 768\) (ResNet38–BERT: 512, BEATs–BERT: 768). In this regime, the feature-level OT block adds only two $d \times d$ float32 matrices $\mathbf{C}_{\text{Feature}}$ and $\mathbf{P}_b$, which together occupy only a few megabytes. Empirically, for BEATs–BERT with batch size 64, the peak GPU memory increases from 20.9 GB (instance-level baseline) to 21.6 GB with DART+RAM, i.e., less than 4% relative overhead.
>
> Runtime. On the same setting, we did not observe any systematic slowdown when enabling DART+RAM. For example, on AudioCaps (BEATs–BERT, batch size 64), one representative epoch without RAM took about 5.4 minutes, while the corresponding run with DART+RAM took about 3.9 minutes. On Clotho, the per-epoch times were about 4.8 minutes without RAM and 3.1 minutes with DART+RAM. These differences are within normal run-to-run variability due to data loading and I/O, and indicate that the feature-level OT block adds only modest overhead compared to the base encoders.
>
>
> **Q6**: In Eq. (10), how to calculate kurt?
>
> **R6**: Thank you for pointing out this missing detail. We have added an explicit definition of the kurtosis term in the revised version.
> The kurtosis for channel $j$ is then
>
> $$
> \text{kurt}_j = \frac{1}{k} \sum_{i=1}^k \left( \frac{z_{ij} - \mu_j}{\sigma_j} \right)^4 - 3.
> $$
>
> This quantity is large when the channel distribution is heavy-tailed and dominated by rare extreme values, which is exactly the type of behavior we want RAM to penalize. We now describe this computation in the main text and give the full formula in the appendix for completeness.

---

### Official Review · Reviewer_kiaK · 2025-10-31

**Soundness:** 2
**Presentation:** 3
**Contribution:** 2
**Rating:** 6
**Confidence:** 3

**Summary:**

This paper presents a dual-level alignment via robust transport (DART) method for audio-text retrieval. The proposed method has been explained in detail and experiments have been condudcted for evaluation.

**Strengths:**

This proposed method combines both instance-level alignment and feature-level regularization for cross-modal retrieval. Plenty of experiments have been condcuted to answer four key questions.

**Weaknesses:**

The novelty of the proposed method needs more clarification. I don't think the introduction section gives a clear explanation on the connection between the proposed method and existing ones. Accroding to the Related Work section, channel-level considerations have been made in previous methods. Thus, the contribution of this paper would be not so significant if it only combines both instance and feature-level alignments. Besides, the paper writing still needs improvement. For example, there are ?? at line137/142.

**Questions:**

According to Table 1, using Beats as audio encoder achieves better performance. Please explain why this configuration was not adopted in following experiments.

---

> ### Author Response · Authors · 2025-12-03
>
> **Q1**: The novelty of the proposed method needs more clarification. I do not think the introduction section gives a clear explanation of the connection between the proposed method and existing ones. According to the Related Work section, channel-level considerations have been made in previous methods. Thus, the contribution of this paper would not be very significant if it only combines both instance and feature-level alignments.
>
> **R1**: We appreciate this comment and have revised both the Introduction and Related Work to clarify the conceptual and technical novelty of DART beyond prior channel-level methods.
>
> First, the core limitation of existing alignment methods is structural: they rely on an instance-level loss whose concentration bound is fundamentally controlled by the maximum pairwise distance $D_\text{max}$ within a mini-batch (Theorem 1). This makes the objective highly susceptible to outliers and noisy channels. Prior channel-level reweighting (e.g., Luong et al.[1] ) does not resolve this structural issue, because it still operates within the instance-distance framework: it only changes how the instance distance $d(x_i,y_j)$ is computed, without altering the fundamental dependence on $D_\text{max}$.
>
> This motivates DART to step outside the instance-distance framework and introduce a separate feature-level OT objective. By focusing on aligning the statistical distributions of the feature channels across the entire batch, the proposed feature-level loss is governed by an aggregate quantity (the Frobenius norm of the transport plan, Theorem 2) rather than a single extremal pair, so that the overall training signal is no longer dominated solely by a single worst-case distance.
>
> Second, the proposed RAM implements mass modulation in the transport problem rather than diagonal feature reweighting, making it a fundamentally different channel mechanism compared to prior methods. RAM operates inside the feature-level OT objective: the corr–var–kurt scores are normalized into marginal distributions that determine how much transport mass each channel is allowed to send or receive. Instead of merely scaling feature amplitudes, RAM directly limits the contribution of unreliable channels to the global transport and lets reliable semantic channels carry most of the mass. This formulation also naturally supports flexible cross-channel couplings ($j:k$) through the transport plan $\mathbf{P}_{\text{Feature}}$, which diagonal reweighting cannot express.
>
> **Q2**: Besides, the paper writing still needs improvement. For example, there are ?? at line137/142.
>
> **R2**: Thank you for pointing this out. We have carefully proofread the manuscript and removed the “??” placeholders at lines 137/142, along with several other minor typos.
>
> **Q3**: According to Table 1, using Beats as audio encoder achieves better performance. Please explain why this configuration was not adopted in following experiments.
>
> **R3**: Regarding the encoder choice, our main experiments adopt the ResNet38–BERT configuration in order to keep the setup fully consistent with the inverse OT baseline of Luong et al., so that any performance differences can be attributed to the proposed DART+RAM rather than changes in backbone capacity. Following your suggestion, we have additionally run DART and DART+RAM with the BEATs–BERT encoder. The corresponding results are now reported in Table R3, and they show that DART+RAM also brings consistent gains on top of the stronger BEATs–BERT backbone.
>
> | Dataset   | noise_p | Method              | A→T R@1 | A→T R@5 | A→T R@10 | T→A R@1 | T→A R@5 | T→A R@10 |
> |-----------|---------|---------------------|--------:|--------:|---------:|--------:|--------:|---------:|
> | AudioCaps | 0.2     | DART                |   54.0  |   84.0  |    92.7  |   67.2  |   90.9  |    96.3  |
> | AudioCaps | 0.2     | SOTA (Chen et al.)  |   50.9  |   81.2  |    88.0  |   66.1  |   89.3  |    96.7  |
> | AudioCaps | 0.4     | DART                |   51.1  |   82.1  |    90.8  |   64.2  |   89.8  |    95.7  |
> | AudioCaps | 0.4     | SOTA (Chen et al.)  |   49.2  |   76.7  |    82.4  |   62.8  |   87.9  |    95.1  |
> | Clotho    | 0.2     | DART                |   26.3  |   54.2  |    66.4  |   33.0  |   59.1  |    72.9  |
> | Clotho    | 0.2     | SOTA (Chen et al.)  |   24.2  |   50.5  |    62.3  |   32.1  |   57.0  |    70.1  |
> | Clotho    | 0.4     | DART                |   24.4  |   50.3  |    62.6  |   30.1  |   56.5  |    68.6  |
> | Clotho    | 0.4     | SOTA (Chen et al.)  |   21.8  |   48.2  |    60.6  |   27.4  |   52.7  |    66.5  |
>
> [1] Luong, M., Nguyen, K., Ho, N., Haf, R., Phung, D., & Qu, L. (2024). Revisiting deep audio-text retrieval through the lens of transportation. arXiv preprint arXiv:2405.10084

---

### Official Review · Reviewer_LD5Q · 2025-11-01

**Soundness:** 3
**Presentation:** 3
**Contribution:** 2
**Rating:** 6
**Confidence:** 3

**Summary:**

The authors propose a dual-level optimal transport framework that integrates instance-level alignment with feature-level regularization via Unbalanced Wasserstein Distance and Reliability-Aware Marginals, achieving more robust and stable cross-modal (audio–text) retrieval under small-batch and noisy-label conditions.

**Strengths:**

1. The authors propose a novel and reasonable solution that incorporates instance-level inverse optimal transport and feature-level unbalanced Wasserstein regularization within a dual-level optimal transport framework.
2. Introducing reliability-aware marginals (RAM) to reweight feature channels based on cross-modal statistics is intuitive and effective.

**Weaknesses:**

1. Lack of Deeper Ablation for RAM Components: The existing ablation experiment table (1) only compares "DART w/ RAM" with "DART w/o RAM". I think more fine-grained ablation experiments should be provided. For example: (1) Use only corr, (2) Use only corr-var, (3) use other combinations (such as weighted sum). Without these experiments, we cannot determine if components like kurtosis are necessary or if the current combination is optimal.
2. Computational scalability concerns: The feature-level OT module introduces a cost matrix of size *d×d*, leading to quadratic complexity in feature dimensionality.   The paper does not provide sufficient discussion on scalability to high-dimensional encoders such as CLIP or BEATs.

**Questions:**

1. As mentioned in Weakness 1, the reliability score formula corr - var - kurt seems heuristic. Can the authors provide more theoretical or empirical evidence for choosing this specific combination?
2. As noted in Weakness 2, the O(d^2) complexity is a potential threat. Have the authors considered methods to mitigate this issue in high-dimensional settings (e.g., d > 2048)?

---

> ### Author Response · Authors · 2025-12-03
>
> **Q1**: Lack of Deeper Ablation for RAM Components / Heuristic Nature of Combination.
>
> **R1** Thank you for this suggestion. We agree that a more fine-grained ablation is important for understanding the design of RAM. In the revision we now
> (i) decompose RAM into the core variants uniform, corr, emavar, kurt, and full RAM, and
> (ii) further evaluate correlation-based hybrids such as corr-gap and corr-burt. The complete results are reported in Tables R1–R2, where we show that:
>
> • corr alone is unstable and underperforms the uniform baseline on mean R@1;
>
> • variance-only (emavar) and kurtosis-only (kurt) each improve over uniform;
>
> • naïve “corr + X” variants (corr-gap, corr-burt) are still weaker than emavar/kurt and the full RAM;
>
> • the full RAM (corr + EMA variance + kurtosis) yields the best mean R@1.
>
> For clarity, we reproduce the main tables below.
> Table R1: Core RAM variants on AudioCaps (ResNet38–BERT, batch size 64).
> | Marginal Design        | A→T R@1 (%) | A→T R@10 (%) | T→A R@1 (%) | T→A R@10 (%) | Mean R@1 (%) |
> |------------------------|------------:|-------------:|------------:|-------------:|-------------:|
> | **uniform (w/o RAM)**  | 51.52       | 90.80        | 38.31       | 85.77        | 44.92        |
> | **corr (Correlation)** | 50.05       | 90.60        | **38.64**       | 85.22        | 44.35        |
> | **emavar (EMA var)**| 51.83       | 90.49        | 38.52       | 85.56        | 45.18        |
> | **kurt (kurtosis)**    | 51.93       | 90.60        | **38.64**       | 85.74        | 45.29        |
> | **RAM (Full)**         | **52.56**   | 90.60        | 38.54       | 85.56        | **45.55**    |
>
> Table R2: Corr-based RAM variants on AudioCaps (ResNet38–BERT, batch size 64). We report R@1 to highlight the main trends.
>
> | Marginal Design | A→T R@1 (%) | T→A R@1 (%) | Mean R@1 (%) |
> |-----------------|------------:|------------:|-------------:|
> | corr-gap        | 51.83       | 38.12       | 44.97        |
> | corr-burt       | 50.99       | 38.75       | 44.87        |
>
>
> **Q2**: The feature-level OT module introduces a cost matrix of size $d \times d$, leading to $O(d^2)$ complexity. Have the authors considered mitigation methods for high-dimensional settings?
>
> **R2**: Thank you for raising this concern. Regarding the $O(d^2)$ term, our feature-level OT block only adds a single $d \times d$ cost matrix (and its transport plan) on top of the instance-level baselines. As discussed in the main paper (paragraph "DART introduces negligible GPU memory overhead compared to instance-level baselines"), in our experiments the feature dimension is at most $d=768$ (ResNet38–BERT uses $d=512$, BEATs–BERT uses $d=768$). In this regime, the two $d \times d$ float32 matrices for $\mathbf{C}_{\text{Feature}}$ and $\mathbf{P}_b$ occupy only a few MB. Even when considering $d=2048$, the two matrices would require approximately 32 MB of memory ($2 \times 2048^2 \times 4$ bytes), which is still far less demanding than the $O(N^3 \log N)$ complexity associated with increasing the batch size $N$ in instance-level OT. Empirically, enabling DART+RAM increases peak memory from 20.9 GB (instance-level baseline) to 21.6 GB (with feature-level OT) when $d=768$.
>
> For the theoretical threat posed by extremely high-dimensional encoders ($d > 2048$), we propose two clear mitigation strategies to ensure scalability: (1) Dimensionality Projection, where features are projected down to a lower dimension ($d' \le 1024$) via a lightweight linear layer before OT calculation, thereby reducing complexity to $O(d'^2)$ while retaining semantic alignment; and (2) Low-Rank Approximation of the cost matrix using established techniques, such as Nyström methods, to achieve sub-quadratic complexity. We believe that a detailed empirical study of these scaling strategies is an interesting direction for future work, and we thank the reviewer for this suggestion.

---

### Official Review · Reviewer_P5Cb · 2025-11-02

**Soundness:** 2
**Presentation:** 1
**Contribution:** 2
**Rating:** 6
**Confidence:** 4

**Summary:**

The paper proposes DART, a framework that enhances cross-modal retrieval by combining instance-level alignment with feature-level regularization based on the Unbalanced Wasserstein Distance. By reweighting embedding channels according to cross-modal consistency, DART suppresses noisy dimensions and stabilizes learning under small batches and scarce labels.

**Strengths:**

* The paper goes beyond conventional instance-level contrastive alignment by introducing a feature-level regularization mechanism based on UOT and IOT. This dual-level design elegantly captures both instance-wise and dimension-wise consistency, addressing the long-standing assumption that all embedding dimensions are equally informative.

* The authors provide clear theoretical analysis showing how instance-level alignment objectives scale with the maximum distance among aligned pairs, while feature-level regularization scales with the Frobenius norm of the transport plan. This theoretical distinction explains DART’s robustness to noise and its improved generalization under small batch regimes.

**Weaknesses:**

* The ablation study is poorly written so that it is not clear how each critical design choice is justified.

* In the Introduction, the authors claim that ‘noisy channels tend to incur large transport costs.’ However, no empirical evidence is provided to support this claim, and it remains unclear whether the proposed feature alignment method effectively addresses this issue.

**Questions:**

The ablation study needs a major revision.

---

> ### Author Response · Authors · 2025-12-03
>
> **Q1**: The ablation study is poorly written so that it is not clear how each critical design choice is justified. The ablation study needs a major revision.
>
> **R1**: Thank you for pointing this out. In the revision, we have reorganized and extended the ablation study so that the construction of RAM and its components is explicit and easy to follow. We first decompose the reliability-aware marginal into the following core variants:
>
> • uniform: all channels treated equally (no RAM);
>
> • corr: weights based only on cross-modal correlation;
>
> • emavar: an EMA-based variance term that down-weights unstable channels;
>
> • kurt: a kurtosis-based variant that penalizes heavy-tailed, outlier-dominated channels;
>
> • RAM: the full combination of correlation, variance, and kurtosis into a single reliability score $r_j$.
>
> All variants are evaluated on the AudioCaps dataset with batch size 64 using the ResNet38–BERT encoders as in Table 1. Their retrieval performance is summarized in Table R1 (to be added in the appendix), where we report R@1 and R@10 for audio→text (A2T) and text→audio (T2A), as well as the average R@1 across both directions.
>
> Table R1: Core RAM variants on AudioCaps (ResNet38–BERT, batch size 64).
> | Marginal Design        | A→T R@1 (%) | A→T R@10 (%) | T→A R@1 (%) | T→A R@10 (%) | Mean R@1 (%) |
> |------------------------|------------:|-------------:|------------:|-------------:|-------------:|
> | **uniform (w/o RAM)**  | 51.52       | 90.80        | 38.31       | 85.77        | 44.92        |
> | **corr (Correlation)** | 50.05       | 90.60        | **38.64**       | 85.22        | 44.35        |
> | **emavar (EMA var)**| 51.83       | 90.49        | 38.52       | 85.56        | 45.18        |
> | **kurt (kurtosis)**    | 51.93       | 90.60        | **38.64**       | 85.74        | 45.29        |
> | **RAM (Full)**         | **52.56**   | 90.60        | 38.54       | 85.56        | **45.55**    |
>
> From Table R1 we observe that:
>
> (1) **Correlation Alone is Unstable.**
> The *corr* variant slightly improves T→A R@1 over the uniform baseline (38.64 vs 38.31) but hurts A→T R@1 (50.05 vs 51.52), leading to a lower mean R@1 (44.35 vs 44.92). This empirical finding strongly supports our claim that simple cross-modal correlation is unstable and prone to being dominated by spurious signals in small batches.
>
> (2) **Variance-based stability is necessary.**
> Adding the EMA variance term (*emavar*) consistently improves over uniform in both directions (A→T R@1: 51.83 vs 51.52; T→A R@1: 38.52 vs 38.31), raising the mean R@1 to 45.18%. This supports our claim that down-weighting high-variance (unstable) channels is crucial for robust transport.
>
> (3) **Kurtosis and the full RAM give the best overall robustness.**
> The *kurt* variant further improves the mean R@1 to 45.29%, and the full RAM combining corr+var+kurt achieves the best mean R@1 of **45.55%**, with the highest A→T R@1 (52.56) while keeping T→A R@1 competitive. This suggests that kurtosis provides an extra safeguard against heavy-tailed, outlier-dominated channels, complementing the variance term.
>
> **Further variants with correlation-based hybrids.**
> For completeness, we also evaluated two additional hybrids, corr-gap (correlation + gap-style variance) and corr-burt (correlation + kurtosis), reported in Table R2. Both variants slightly improve over pure corr, confirming that adding stability statistics is helpful; however, their mean R@1 (44.98% and 44.87%) remains below the variance-only emavar (45.18%), the kurtosis-only kurt (45.29%), and the full RAM (45.55%). These negative results show that naïvely combining correlation with a single extra statistic does not yield a robust marginal, and that our final RAM design — which integrates EMA variance and kurtosis in a unified reliability score, instead of simply “corr + X” — is necessary to obtain consistent gains.
>
> Table R2: Corr-based RAM variants on AudioCaps (ResNet38–BERT, batch size 64). We report R@1 to highlight the main trends.
>
> | Marginal Design | A→T R@1 (%) | T→A R@1 (%) | Mean R@1 (%) |
> |-----------------|------------:|------------:|-------------:|
> | corr-gap        | 51.83       | 38.12       | 44.97        |
> | corr-burt       | 50.99       | 38.75       | 44.87        |

---

> ### Author Response · Authors · 2025-12-03
>
> **Q2**: In the Introduction, the authors claim that ‘noisy channels tend to incur large transport costs.’ However, no empirical evidence is provided to support this claim, and it remains unclear whether the proposed feature alignment method effectively addresses this issue.
>
> **R2**: Thank you for raising this point. In the revision, we added two empirical studies (new Figs. 2–3 in Appendix N) that directly support this claim and show how RAM behaves in practice.
>
> 1. **Controlled noise vs. Optimal Transport (OT) cost.** We first design a synthetic experiment where we fix a clean pair of audio–text feature distributions and inject i.i.d. Gaussian noise with standard deviation $\sigma \in$ {$0,0.1,0.2,0.3$} into one modality. For each $\sigma$, we compute (i) the OT cost between the clean features (blue curve) and (ii) the OT cost between the noisy features and the same clean counterpart (orange curve). The resulting plot shows that the noisy OT cost grows monotonically with $\sigma$ and becomes clearly larger than the clean baseline even at moderate noise levels. This directly supports our statement that stronger channel noise systematically leads to larger transport costs.
>
> 2. **Reliability vs. standardized cost under DART.** On a trained DART model, we further compute, for each feature channel $j$, its Reliability Score $r_j$ and the standardized OT cost $\tilde{C}_j$  (which normalizes channel-wise costs to remove scale/batch effects).  Figure 3 plots $\tilde{C}_j$ against $r_j$ for all channels and reports the Pearson correlation $\rho(r, \log \tilde{C}) \approx \mathbf{-0.379}$. Channels with low reliability (highlighted in red) consistently correspond to higher standardized cost, whereas channels with high reliability (highlighted in green) lie in the low-cost region. This negative correlation shows that our reliability-aware feature alignment indeed down-weights high-cost (noisy) channels and concentrates transport mass on stable semantic channels.
>
> Together, these two figures provide the requested empirical evidence: noisy channels do incur larger transport costs, and the proposed RAM mechanism effectively identifies and suppresses their influence in practice.

---

### Author Response · Authors · 2025-12-03

We are thankful to all reviewers for their efforts and constructive feedback. We are encouraged that Reviewers P5Cb, LD5Q, kiaK, and BWQA find our problem setup and direction relevant and interesting. We are particularly pleased that our theory is deemed novel and the paper is clearly written.
Below is a summary of our responses:

* To **Reviewer P5Cb**: We provide empirical evidence supporting the claim that noisy channels incur larger transport costs via a controlled noise-injection experiment (new Fig.2 in appendix N), where we compare OT costs between clean features and their synthetically noised counterparts and also report the cost increase $\Delta C = C_{\text{noisy}}- C_{\text{clean}}$ as the noisy level grows. We further demonstrate a clear negative correlation between the proposed channel reliability $r$ and standard transport cost (new Fig. 3, Appendix N), confirming that our feature-level alignment effectively suppresses noisy channels and emphasizes reliable ones.

* To **Reviewer LD5Q**: We add fine-grained ablations for the RAM components (corr / corr–var / corr–var–kurt and weighted variants) and provide a detailed discussion of the computational complexity of the feature-level OT, including practical scaling strategies for higher-dimensional encoders.

* To **Reviewer kiaK**: We clarify the novelty of DART relative to prior channel-weighting methods by explicitly contrasting our dual-level OT formulation and reliability-weighted marginals with existing per-dimension reweighting schemes, and we explain our encoder choices and BEATs results in more detail.

* To **Reviewer BWQA**: We have substantially revised the writing for clarity (fixing missing references and notation). We now make it explicit that the limitation discussed in Section 2.3 is already supported by our theory in Section 4: Theorem 1 provides a concentration bound where the deviation of the instance-level IOT loss is controlled by the worst-case alignment distance $D_{\max}$, which is precisely inflated by noisy or unstable channels. In addition, we complement this analysis with new empirical evidence (Appendix N, Figs. 2-3) that demonstrates how injected noise increases OT cost and how our reliability-aware formulation suppresses high-cost channels in practice, and we further expand the RAM ablations and runtime/memory comparisons.

---

### Meta-Review · Area_Chair_1kiA · 2026-01-10

**Summary:**

This paper receives the scores of 6, 6, 6, and 4 (average 5.5). The weaknesses pointed out by the reviewers are:
- Insufficient analysis of the ablation study for RAM components, and the absence of runtime/memory comparisons.
- A perceived lack of empirical evidence to support claims regarding noisy channels and the limitations of instance-level IOT.
- General writing quality issues, including minor typos and lack of clarity.

**Reviewer Concerns:**

In the feedback, the authors provide more ablation studies with detailed experiments, adding empirical evidence to support claims about noisy channels, and providing runtime/memory comparisons with scalability mitigations. They also clarified the paper's novelty in relation to prior work and improved the overall writing quality. These concerns are well addressed.

**Reviewer Scores:**

Given the author's detailed response to the reviewers' questions, I think all reviewers will either maintain or even increase their scores slightly.

---

### Decision · Program_Chairs · 2026-01-26

Accept (Poster)